# Holistic protean block for long-range DNA sequence modeling

## Abstract

Modeling DNA sequences, which are defined by a complex interplay of local motifs, long-range dependencies, and periodic patterns, is a fundamental challenge in computational biology. Existing foundation models based on CNNs, Transformers, and SSMs are constrained by static or time-domain-only signal processing operations, which limit the flexibility and multi-domain perspective needed to fully capture the diverse features of DNA. Here, we introduce the **Holistic Protean Block (HPB)**, a novel scalable architecture that achieves multi-level plasticity through three synergistic layers. Its Locus Plasticity Layer (LPL) provides token-level plasticity by employing token-specific convolution operations, allowing it to precisely model fine-grained, local patterns. Its Domain Plasticity Layer (DPL) establishes perspective-level plasticity by concurrently modeling both sequential (time) and spectral (frequency) features, enabling it to form multi-domain, global representations. Its Saliency Plasticity Layer (SPL) realizes information flow plasticity by learning saliency scores along dual axes, thereby permitting it to focus information flow on the most critical features. These layers work in tandem, extracting a holistic representation of diverse genomic patterns by adaptively reshaping their computational strategy. The DNA model constructed with HPB (**HPB-DNA**) not only achieves state-of-the-art performance on various genomic benchmarks with a quasi-linear complexity, but is also validated by in-depth model analyses, which collectively establish the HPB as a more powerful and principled paradigm for DNA sequence modeling. Code will be available upon acceptance.

## 1 Introduction

Genomic sequence analysis presents a significant challenge due to the extraordinary length and informational complexity of DNA. The advancement of computational models, however, offers a powerful solution. Leveraging their efficiency and precision, these models are uniquely suited to process vast and intricate biological data. Consequently, applying computational methods to genomic analysis has become an increasingly prevalent and fruitful approach, showing great promise in deciphering the complexities of the genome (Bellot et al., 2018; Amin et al., 2018; Eraslan et al., 2019; Zou et al., 2019; Tian et al., 2019; Wang et al., 2020b; Talukder et al., 2021).

DNA sequences exhibit a profound informational diversity, characterized by sparse yet critical features, such as local motifs (Wasserman & Sandelin, 2004), long-range dependencies (Schoenfelder & Fraser, 2019), and periodic patterns (Brogaard et al., 2012). However, existing DNA foundation models are constructed upon common architectures, such as Convolutional Neural Network (CNN) (Zhou & Troyanskaya, 2015; Bo et al., 2025), Transformer (Zhou et al., 2024a; Dalla-Torre et al., 2025), and State Space Model (SSM) (Schiff et al., 2024). These architectures rely on basic modules with static or single-domain operations, which limit their architectural flexibility in capturing diverse features inherent in DNA sequences. CNNs, while effective at identifying local motifs, struggle to capture long-range dependencies due to their static kernels and limited receptive fields (Kelley et al., 2016; Liu et al., 2022; Yu & Wang, 2025). Transformers (Vaswani et al., 2017) excel at modeling long-range interactions through the attention mechanism, but lack the strong inductive biases for local patterns inherent to CNNs, making it more challenging to effectively model local features for DNA sequences (Avsec et al., 2021). Moreover, their quadratic complexity is computationally prohibitive for genome-scale sequences. Although SSM models (e.g., Mamba)

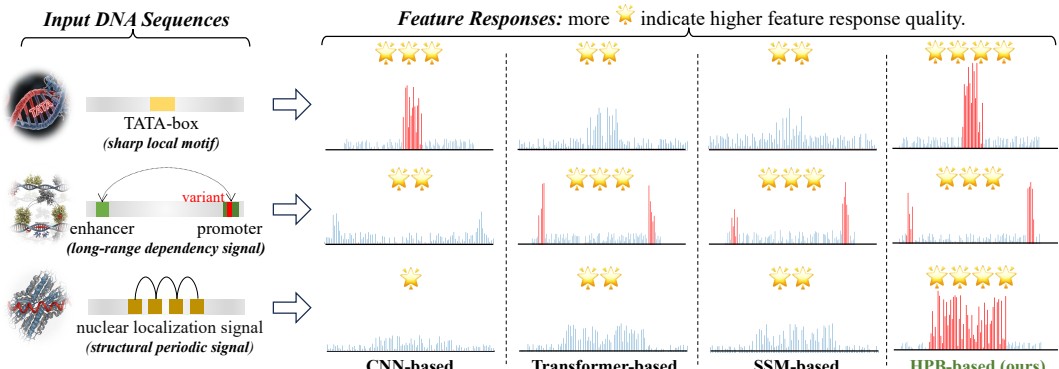

Figure 1: **Comparison of feature responses to diverse DNA patterns across different models.** CNN-based models are adept at local feature extraction but are structurally constrained in capturing long-range and periodic patterns. Transformer-based and SSM-based models excel at capturing global dependencies but show limited efficacy in modeling local motifs and frequency-sensitive signals. In contrast, the HPB-based model utilizes a **'protean'**, multi-level plasticity mechanism to achieve a holistic representation that integrates local, global, and frequency-sensitive features.

achieve linear complexity, their lossy state compression may lose fine-grained details (Gu & Dao, 2023). Other works (Avsec et al., 2021; Yang et al., 2025) adopt hybrid architectures as a pragmatic compromise, but still fall short of fundamentally addressing DNA's inherent multi-scale complexity.

A deeper, universal limitation is that these models operate exclusively in the time domain and thus lack the intrinsic mechanisms to directly model crucial frequency-sensitive features. Consequently, vital periodic signals (including codon periodicity (Yin & Yau, 2007), nucleosome positioning signals (Peckham et al., 2007; Brogaard et al., 2012), and tandem repeats (Weber & Wong, 1993; Benson, 1999)) remain poorly represented by current architectures, as illustrated in Fig. 1.

To address these limitations, we propose the **Holistic Protean Block (HPB)**, a novel scalable building block unified by the principle of multi-level plasticity (Lyle et al., 2023; Dohare et al., 2024). The block begins with the Locus Plasticity Layer (LPL), which dynamically extracts robust local features from DNA embeddings. It achieves this by using varying-sized convolutional kernel bases to capture multi-scale meta features (basic features used for token-level feature construction), and generating token-specific weights through a conv layer to combine those meta features into fine-detailed representations. These representations are then processed in parallel by the Domain Plasticity Layer (DPL) and the Saliency Plasticity Layer (SPL). The DPL employs a global convolution and a wavelet transform to fuse sequential patterns with spectral features, allowing it to model both global dependencies and frequency-sensitive representations. Simultaneously, the SPL computes saliency scores across both the channel and position axes, identifying the most informative elements. Finally, the DPL's time-frequency representations and the SPL's saliency scores are combined via element-wise multiplication, which selectively amplifies critical information while attenuating noise. Through the synergistic interplay of these three layers, the HPB organically integrates local-global, time-frequency domain information, achieving a holistic (Bertsimas et al., 2024; Yang et al., 2020) capture of features and dependencies within DNA sequences, as shown in Fig. 1. Furthermore, as a fundamental building block, HPB can be stacked to construct various-sized DNA foundation models, termed **HPB-DNA**.

Across a wide array of benchmarks testing both short- and long-range sequence dependencies, our method establishes a new state-of-the-art, outperforming previous methods by a notable margin. Thorough ablation studies and model analyses further validate the advantages of our approach, demonstrating its strong capabilities for modeling the complex language of DNA.

The contributions of this work are listed below:

- We introduce the Holistic Protean Block (HPB), a novel and scalable building block for DNA sequence analysis. Its design, centered on multi-level plasticity, allows it to dynamically adjust its computational strategy to capture the diverse patterns inherent in DNA data.

- We design the Locus Plasticity Layer (LPL) to capture fine-grained local patterns through a token-specific extraction mechanism; the Domain Plasticity Layer (DPL) to jointly model and fuse time-frequency representations using a global convolution and a wavelet transform; and the Saliency Plasticity Layer (SPL) to learn dynamic saliency scores along channel and position axes for feature refining.
- The HPB-based DNA model (HPB-DNA) achieves state-of-the-art performance across various benchmarks while remaining highly efficient in terms of both parameter count and computational complexity, which underscores its strong potential as a leading paradigm in DNA sequence modeling.

## 2 RELATED WORK

**DNA Sequence Analysis**  Analyzing and interpreting the vast information encoded within DNA sequences is a fundamental challenge in modern genomics. This genetic information, spanning both coding and non-coding regions, dictates everything from protein synthesis to the complex orchestration of gene expression, making it critical for understanding health and disease. Traditional bioinformatic methods, which are largely statistical in nature, often fail to capture the intricate, context-dependent patterns in DNA sequences. Deep learning methods, such as CNN (He et al., 2016), Transformer (Vaswani et al., 2017), and SSM (Gu & Dao, 2023), have driven a paradigm shift by learning hierarchical features directly from raw data, overcoming these limitations.

**Time and Frequency Modeling**  In the field of signal processing, the analysis of one-dimensional data has historically been divided into two core perspectives: time-domain and frequency-domain. Time-domain analysis operates directly on the sequence, using methods like digital filters (e.g., (Schafer, 2011)) and autoregressive models to capture local behavior. The central idea of these methods is to understand the signal's dynamics by analyzing its temporal dependencies. Frequency-domain analysis focuses on the signal's global and periodic structures; the FFT (Nussbaumer, 1981) identifies spectral components but loses temporal information. The STFT offers a time-frequency compromise via spectrograms, while the Wavelet Transform (Zhang, 2019) provides a flexible multi-resolution analysis ideal for non-stationary signals. These classic paradigms inspire modern deep learning: CNNs act as learnable time-domain filters, while model like Hyena (Nguyen et al., 2023) leverage FFT for efficient long convolutions.

**CNN-based DNA models**  As a foundational deep learning architecture in genomics, Convolutional Neural Networks (CNNs) remain a cornerstone of the field. Early models like DeepBind (Alipanahi et al., 2015) applied CNNs to predict protein-binding specificities, while DeepSEA (Zhou & Troyanskaya, 2015) predicted the functional effects of non-coding variants. More recently, architectures have adapted CNNs to capture long-range dependencies. CondConv (Yang et al., 2019) introduced dynamic convolutions to enhance flexibility; however, its dynamic adaptation is limited to the sample level. Hyena (Nguyen et al., 2023), for example, employed long convolutions to model million-token sequences, while ConvNova (Bo et al., 2025) incorporated advanced features like dilated and gated convolutions in a dual-branch framework to achieve a wider receptive field.

**Transformer-based DNA models**  The introduction of the Transformer architecture marked a pivotal moment in genomics, reframing sequence analysis by treating DNA as a language. DNABERT (Ji et al., 2021) pioneered this approach by introducing a k-mer tokenization strategy. Subsequent iterations improved upon this foundation: DNABERT-2 (Zhou et al., 2024a) enhanced computational efficiency with Byte Pair Encoding (BPE), while DNABERT-S (Zhou et al., 2024b) employed specialized contrastive learning for species-aware embeddings. This foundation model concept was significantly scaled up with the Nucleotide Transformer (NT) (Dalla-Torre et al., 2025), featuring models with up to 2.5 billion parameters. Hybrid architectures have also emerged, such as Enformer (Avsec et al., 2021), which integrated convolutional layers with an attention mechanism to predict gene expression from DNA sequences. Concurrently, models like Space (Yang et al., 2025) utilized large Mixture-of-Experts (MoE) (Yuksel et al., 2012; Masoudnia & Ebrahimpour, 2014; Lin et al., 2024) to learn DNA representations via supervised training on genomic profiles.

**SSM-based DNA models**  State Space Models (SSMs) (Hamilton, 1994; Paninski et al., 2010), particularly the Mamba architecture, have recently emerged as a powerful and efficient alternative for DNA modeling. Caduceus (Schiff et al., 2024) introduced the MambaDNA block, which enforces reverse-complement (RC) equivariance, a critical inductive bias for genomics. Among early state

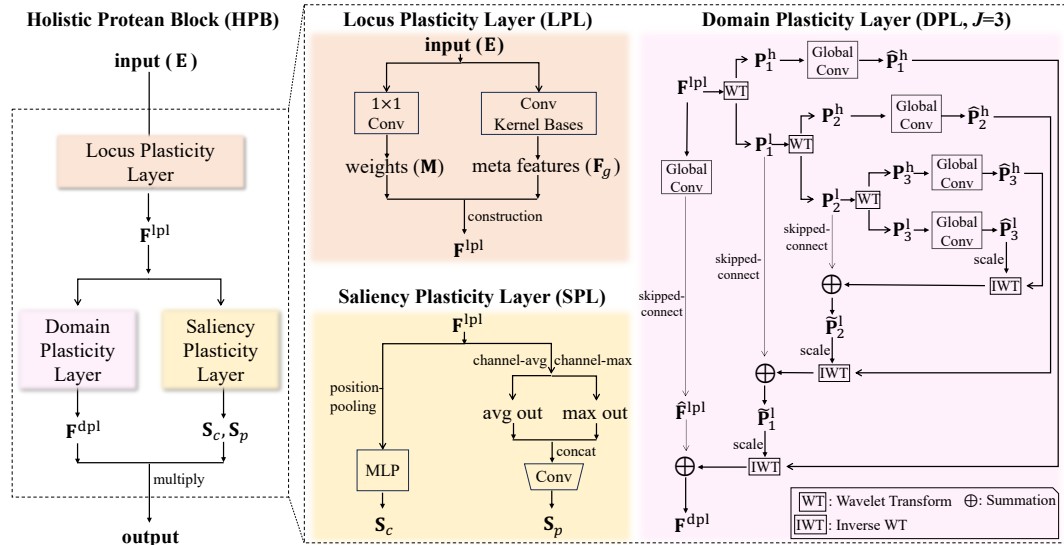

Figure 2: **The architecture of the HPB, which consists of the LPL, the DPL, and the SPL.** The LPL extracts features at each location by combining input-aware, token-specific coefficients with meta-features generated by the conv kernel bases. The DPL models global and cross-domain patterns by utilizing a global conv and the wavelet transform. Finally, the SPL generates dynamic saliency scores along channel and position axes to amplify critical information and suppress noise.

transition models, DanQ (Quang & Xie, 2016) was an effective architecture that combined a bi-directional Long Short-Term Memory (BiLSTM) network with a CNN for DNA analysis. More recently, Bio-xLSTM (Schmidinger et al., 2025) adapted the xLSTM architecture for DNA and protein sequence modeling, achieving high computational efficiency and accuracy.

Despite significant progress, key challenges remain unresolved across these architectures. While CNN-based models are efficient, their ability to model long-range dependencies remains limited. Transformer-based models excel at modeling global dependencies, but they are less effective at directly capturing local features and are hampered by a quadratic computational complexity that is prohibitive for ultra-long DNA sequences. SSM-based models achieve linear complexity but risk losing fine-grained details by compressing information into a finite state. More critically, all these methods operate almost exclusively in the time domain. This shared limitation means they lack a dedicated mechanism to directly model frequency-sensitive patterns, such as nucleosome periodicity, which are crucial in genomics and cannot be effectively leveraged by existing models.

## 3 METHODOLOGY

In this section, we detail the architecture and application of our proposed Holistic Protean Block (HPB). We first present the core design of the HPB, detailing the implementation of its key components. We then describe how to utilize HPB to construct DNA foundation models of varying scales. Finally, we outline the procedures for pre-training and fine-tuning the HPB-based DNA models.

### 3.1 HOLISTIC PROTEAN BLOCK

As shown in Fig. 2(left), the HPB consists of the Locus Plasticity Layer (LPL), the Domain Plasticity Layer (DPL), and the Saliency Plasticity Layer (SPL). The LPL first processes an input DNA embedding ($\mathbf{E}$) with token-specific feature extraction, yielding a contextually-rich output $\mathbf{F}^{\text{lpl}}$. This output is then fed in parallel to the DPL and SPL. The DPL produces unified time-frequency features ($\mathbf{F}^{\text{dpl}}$), while the SPL computes saliency scores across the channel and position axes. The saliency scores and the time-frequency features are combined to obtain the HPB's output.

### 3.1.1 LOCUS PLASTICITY LAYER

As illustrated in Fig. 2(right), the LPL is comprised of two parallel branches: a bank of convolutional kernel bases and a coefficient generation branch that employs a $1 \times 1$ convolution. The kernel bases contain a set of $T$ distinct convolutional kernels, $\mathcal{K} = \{k_0, k_1, \ldots, k_{T-1}\}$ (each with a different size), which are referred to as static "bases". Given an input DNA embedding $\mathbf{E} \in \mathbb{R}^{B \times L \times D}$, where $B$ is the batch size, $L$ is the sequence length and $D$ is the embedding dimension, the kernel bases branch applies each kernel $k_i$ to $\mathbf{E}$ to produce a group of meta features, $\mathbf{F}_g = \{\mathbf{F}_0, \mathbf{F}_1, \ldots, \mathbf{F}_{T-1}\}$, where $\mathbf{F}_i = \text{Conv}(k_i, \mathbf{E})$, $\quad i = 0, \ldots, T-1$. These are basic features used for token-level feature construction. Concurrently, a weight generation branch uses a 1×1 convolution on the input embedding $\mathbf{E}$ to yield a matrix of token-specific weights, $\mathbf{M} \in \mathbb{R}^{L \times T}$. The LPL's output of the $j$-th token, $\mathbf{F}_j^{\text{lpl}}$, is then constructed by using the weights $\mathbf{M}_j = [m_{j,0}, m_{j,1}, \ldots, m_{j,T-1}]$ to create a weighted sum of the corresponding feature vectors $\mathbf{F}_{i,j}$ from the meta features:

$$\mathbf{F}_j^{\text{lpl}} = \sum_{i=0}^{T-1} m_{j,i} \cdot \mathbf{F}_{i,j}, \tag{1}$$

where $\mathbf{F}^{\text{lpl}} \in \mathbb{R}^{B \times L \times D}$. This token-specific construction mechanism provides high flexibility, allowing local context to more effectively guide feature extraction.

### 3.1.2 DOMAIN PLASTICITY LAYER

Following the LPL, the Domain Plasticity Layer (DPL) receives the contextually-rich feature map $\mathbf{F}^{\text{lpl}}$. The primary objective of the DPL is to capture features across different perspectives by concurrently modeling features in both the time and frequency domains, which is achieved by a complementary architecture composed of a Convolution Path and a Wavelet Path (Fig. 2(right)).

**Convolution Path** This path acts as a direct connection to capture global features in the time domain. The input $\mathbf{F}^{\text{lpl}}$ is processed by a Global Convolution operator, yielding the output $\hat{\mathbf{F}}^{\text{lpl}}$. This operator, denoted as GC, performs a non-causal, input-dependent long convolution accelerated by the Fast Fourier Transform (FFT) (Nussbaumer, 1981). The details are provided in Alg. 1.

---

**Algorithm 1** Global Convolution operator (GC)

---

**Input:** Input sequence $\mathbf{F}^{\text{lpl}} \in \mathbb{R}^{B \times L \times D}$.
**Parameters:**
  Learnable decay rates $\delta \in \mathbb{R}^D$ and shift $s \in \mathbb{R}^1$.
  Adaptive Layer Normalization (AdaLN), conditioned by $\mathbf{F}^{\text{lpl}}$.
**Output:** Output sequence $\hat{\mathbf{F}}^{\text{lpl}} \in \mathbb{R}^{B \times L \times D}$.

*Generate symmetric time embedding $t$ to enforce kernel symmetry:*
$t_j \leftarrow 1 - \frac{2}{L-1} \left| j - \frac{L-1}{2} \right| \quad \forall j \in \{0, \ldots, L-1\}$
*Generate complex positional embedding $z$:*
$z_{j,d} \leftarrow \exp(-\text{i} \cdot \frac{2\pi f_d}{L} \cdot j)$  $\quad \triangleright$ i is the imaginary unit, $f_d$ is predefined for the $d$-th dimension.
*Generate data-dependent base filter $h$:*
$h \leftarrow \text{MLP}(\text{AdaLN}(z | \mathbf{F}^{\text{lpl}}))$
*Apply symmetric modulation via $t$ to create the symmetric (non-causal) kernel $p$:*
$p \leftarrow h \odot (\exp(-t \otimes |\delta|) + s)$
$\hat{\mathbf{F}}^{\text{lpl}} \leftarrow \text{iFFT}(\text{FFT}(\mathbf{F}^{\text{lpl}}) \odot \text{FFT}(p))$  $\quad \triangleright \odot$ denotes element-wise multiplication.
*Crop to correct the circular shift and align the input and output:*
$\hat{\mathbf{F}}^{\text{lpl}} \leftarrow \text{crop}(\hat{\mathbf{F}}^{\text{lpl}}, \text{start} = \lfloor L/2 \rfloor, \text{length} = L)$

**return** $\hat{\mathbf{F}}^{\text{lpl}}$

---

**Wavelet Path** This path performs a multi-resolution analysis of the input feature using a discrete Wavelet Transform (WT) (Zhang, 2019). The process involves three main stages: decomposition, frequency-specific processing, and reconstruction. First, the input $\mathbf{F}^{\text{lpl}}$ is iteratively decomposed over $J$ wavelet levels as illustrated for $J = 3$ in Fig. 2(right). At each level $i \in [1, J]$, the low-frequency components from the previous level, $\mathbf{P}_{i-1}^{\text{l}}$ (with $\mathbf{P}_0^{\text{l}} = \mathbf{F}^{\text{lpl}}$), are passed through the WT

to produce new, lower-resolution components $\mathbf{P}_i^l$ and a set of high-frequency components $\mathbf{P}_i^h$:

$$(\mathbf{P}_i^l, \mathbf{P}_i^h) = \text{WT}(\mathbf{P}_{i-1}^l), \quad i = 1, \ldots, J. \tag{2}$$

Then, each set of high-frequency components $\mathbf{P}_i^h$ and the final low-frequency components $\mathbf{P}_J^l$ are processed by the GC operator. This allows the model to learn distinct patterns within different frequency bands:

$$\hat{\mathbf{P}}_i^h = \text{GC}(\mathbf{P}_i^h), \quad i = 1, \ldots, J, \tag{3}$$

$$\hat{\mathbf{P}}_J^l = \text{GC}(\mathbf{P}_J^l). \tag{4}$$

Finally, the signal is reconstructed iteratively, starting from the processed lowest-frequency components, $\hat{\mathbf{P}}_J^l$. This multi-scale feature fusion is defined by the following recursive formula, applied at each level $i$ from $J-1$ down to 1:

$$\tilde{\mathbf{P}}_i^l = \mathbf{P}_i^l + \text{IWT}\left(\gamma_i \cdot \tilde{\mathbf{P}}_{i+1}^l, \quad \hat{\mathbf{P}}_{i+1}^h\right). \tag{5}$$

The recursion is initialized with the base case $\tilde{\mathbf{P}}_J^l = \hat{\mathbf{P}}_J^l$. At each step, the reconstructed signal from the level below ($\tilde{\mathbf{P}}_{i+1}^l$) is scaled by a learnable parameter $\gamma_i$ and then upsampled via the Inverse WT (IWT) using the processed high-frequency components ($\hat{\mathbf{P}}_{i+1}^h$). This result is then added to the original, unprocessed low-frequency components from the decomposition stage ($\mathbf{P}_i^l$) to form the final representations for the current, higher-resolution level, $\tilde{\mathbf{P}}_i^l$. This skip connection preserves the original low-frequency information, preventing the loss of critical patterns. After $J-1$ iterations of Eq. 5, the reconstructed signal $\tilde{\mathbf{P}}_1^l$ is obtained. The final reconstruction step, which combines time- and frequency-sensitive representations, is formulated as: $\mathbf{F}^{dpl} = \hat{\mathbf{F}}^{lpl} + \text{IWT}\left(\gamma_0 \cdot \tilde{\mathbf{P}}_1^l, \hat{\mathbf{P}}_1^h\right)$.

### 3.1.3 SALIENCY PLASTICITY LAYER

Simultaneously, the Saliency Plasticity Layer (SPL) receives the output from the LPL $\mathbf{F}^{lpl} \in \mathbb{R}^{B \times L \times D}$ as its input. The objective of the SPL is to determine the significance of the captured features by generating dynamic, data-dependent weights across both channel and position axes. As illustrated in Fig. 2(right), the SPL is composed of two parallel branches: a Multi-Layer Percep-tron (MLP) branch for channel saliency and a depthwise separable convolution branch for position saliency. In the first branch, the input $\mathbf{F}^{lpl}$ is aggregated along the position axis using average pool-ing to create a channel descriptor vector. This vector is then processed by an MLP to yield the channel saliency scores, $\mathbf{S}_c \in \mathbb{R}^{B \times 1 \times D}$. In parallel, the second branch computes features along the channel axis by calculating both the average and the maximum values of $\mathbf{F}^{lpl}$, which provides a richer position descriptor, enabling the model to capture both sharp motifs via peak activations and broader regulatory regions via average activations. These two values are concatenated and passed through a depthwise separable convolution to perform spatial integration and produce the position saliency scores, $\mathbf{S}_p \in \mathbb{R}^{B \times L \times 1}$. Finally, these two sets of scores, $\mathbf{S}_c$ and $\mathbf{S}_p$, are broadcast to match the dimensions of the features from the DPL, $\mathbf{F}^{dpl}$, and are multiplied element-wise. This opera-tion dynamically refines the features to produce the final output of the HPB. By explicitly modeling both channel-wise ('what') and positional ('where') saliency, SPL provides a more comprehensive feature refinement mechanism than approaches that are solely channel-focused or locally gated.

Table 1: **Performance comparison on Genomic Benchmark.** Top-1 accuracy ($\%, \uparrow$) is reported for the officially released pre-trained models. 'ConvNova' is pre-trained using the official code.

| | HyenaDNA (436k) | CADUCEUS-PH (470k) | ConvNova (386k) | xLSTM-PH (500k) | HPB-DNA (ours) (490k) |
|---|---|---|---|---|---|
| Mouse Enhancers | 78.03±0.19 | 78.60±1.32 | 79.17±0.87 | 78.48±0.94 | **82.38±0.66** |
| Coding vs. Intergenomic | 90.43±0.05 | 91.86±0.28 | 93.35±1.52 | 93.11±1.38 | **94.46±0.47** |
| Human vs. Worm | 95.83±0.16 | 96.67±0.13 | 96.85±0.85 | 96.71±0.65 | **97.19±0.75** |
| Human Enhancers Cohn | 72.46±0.21 | 73.61±0.47 | 73.82±0.56 | 73.88±0.56 | **74.96±0.35** |
| Human Enhancers Ensembl | 89.43±0.02 | 90.11±0.32 | 90.10±0.16 | 90.42±1.04 | **91.08±0.61** |
| Human Regulatory | 88.45±0.08 | 87.82±0.46 | 88.21±1.03 | 87.67±0.31 | **94.20±0.23** |
| Human Nontata Promoters | 94.74±0.20 | 94.72±0.48 | **95.21±1.26** | 95.11±0.86 | 94.78±0.51 |
| Human OCR Ensembl | 78.14±2.01 | 80.17±0.15 | 79.61±0.42 | 81.72±1.16 | **81.95±1.01** |

### 3.2 HPB-DNA MODEL

**Computational Complexity**   We analyze the complexity of the HPB for an input sequence of length $L$. The LPL (convs) and the SPL (MLP and convs) both operate with linear complexity, $O(L)$. The complexity of the DPL is determined by its two parallel paths: a multi-level wavelet analysis ($O(L)$) and an FFT-based global convolution ($O(L\log_2 L)$). Therefore, the global convolution is the computational bottleneck, making the overall complexity of the HPB quasi-linear at $O(L\log_2 L)$.

**Constructing DNA models with HPB**   The HPB is a scalable component for constructing DNA models of varying sizes, termed HPB-DNA. Specifically, the HPB-DNA first embeds a tokenized DNA sequence into a representation, $\mathbf{E}$, which is then processed by an encoder—a stack of HPB layers, each configurable as 'Pre-Norm' or 'Post-Norm', similar to standard Transformer blocks. The final output is then passed to a task-specific prediction head. This modular design allows the model's scale to be easily customized for diverse downstream tasks (see A.1 for further details).

**Pre-training**   We pre-train our models using a Masked Language Modeling (MLM) strategy. Consistent with prior works (Nguyen et al., 2023; Bo et al., 2025), the training corpus is the human reference genome (GRCh38/hg38), from which we randomly sample fixed-length DNA sequences. During pre-training, we randomly mask 15% of the tokens in each sequence and train the model to predict them based on the surrounding bidirectional context. This self-supervised process forces the model to learn the intricate contextual relationships between nucleotides.

**Finetuning**   To adapt the pre-trained model for downstream applications, we append a task-specific prediction head to the encoder output for finetuning. For instance, to fine-tune the model for a DNA sequence classification task, we first aggregate the encoder's final hidden states into a single global vector via average pooling across the sequence length dimension. This global feature vector is then passed through a linear projection layer to produce the final logits for the classification task.

## 4 EXPERIMENTS

In this section, we evaluate our model's performance on tasks involving both short- and long-range DNA sequences. We then delve into the effectiveness and efficiency of our proposed architecture through a series of ablation studies and in-depth model analyses.

Table 2: **Performance Comparison on Nucleotide Transformer Benchmark.** Performance $(\%, \uparrow)$ is reported for the officially released pre-trained models. 'ConvNova' is pre-trained using the officially released code. Metrics vary by task: MCC for histone markers and enhancers, F1-score for promoters and splice site acceptor/donor, and accuracy for splice site all. The best results for models under 5M parameters are bolded, and the best overall results are underlined.

| | >100M Param. | | <5M Param. | | | | |
| | DNABERT-V2 (117M) | NT-V2 (500M) | HyenaDNA (1.6M) | CADUCEUS-PH (1.9M) | ConvNova (1.7M) | xLSTM-PH (2.0M) | HPB-DNA (ours) (1.9M) |
|---|---|---|---|---|---|---|---|
| *Regulatory Annotation* | | | | | | | |
| Promoter all | 97.06±0.11 | 97.61±0.12 | 96.22±0.29 | 96.13±0.10 | 96.83±0.21 | 96.80±0.76 | **97.01±0.71** |
| Non-TATA | 97.18±0.16 | 97.53±0.14 | 95.60±0.25 | 96.03±0.16 | 96.19±0.53 | 96.11±0.62 | **96.88±0.16** |
| TATA | 95.50±0.39 | 96.35±1.06 | 95.48±0.24 | 95.46±0.36 | 96.50±0.67 | 95.41±0.58 | **96.59±0.42** |
| Enhancer | 55.81±1.60 | 56.62±3.20 | 53.82±0.40 | 50.68±2.41 | 57.78±0.29 | 55.01±0.33 | **58.69±0.49** |
| Enhancer types | 43.78±3.01 | 45.95±2.30 | 48.49±0.85 | 39.21±1.85 | 49.89±0.60 | 47.01±0.52 | **50.87±0.22** |
| *Splice Site Annotation* | | | | | | | |
| Acceptor | 96.46±0.35 | 98.64±0.22 | 96.39±0.74 | 96.54±0.28 | 96.31±0.47 | 95.54±1.11 | **97.16±0.11** |
| Donor | 96.11±0.23 | 98.86±0.18 | 95.26±0.63 | 94.68±0.92 | 96.80±0.55 | 95.69±1.05 | **97.79±0.25** |
| All | 93.70±0.40 | 98.37±0.14 | 95.52±0.97 | 95.13±0.32 | 96.36±0.19 | 95.90±0.58 | **97.12±0.51** |
| *Histone Markers* | | | | | | | |
| H3 | 76.38±2.73 | 79.14±1.21 | 79.89±0.61 | 78.45±0.67 | 81.62±1.23 | 80.97±0.59 | **82.03±0.38** |
| H3K4me1 | 50.75±1.20 | 54.21±0.74 | 51.81±0.79 | 47.20±0.89 | 53.87±1.38 | 52.66±1.25 | **55.39±0.56** |
| H3K4me2 | 31.12±2.63 | 32.12±2.99 | 46.87±1.08 | 45.67±1.35 | 53.39±0.65 | 51.29±0.81 | **53.89±0.23** |
| H3K4me3 | 36.59±1.74 | 40.25±3.09 | 49.29±0.62 | 47.92±1.67 | **58.21±1.59** | 53.80±0.66 | 54.86±0.68 |
| H3K9ac | 53.68±0.90 | 56.62±0.88 | 59.02±0.54 | 60.39±0.92 | 64.12±0.87 | 62.91±0.37 | **64.50±1.19** |
| H3K14ac | 50.20±1.14 | 54.07±1.67 | 55.70±0.39 | 58.54±1.02 | 59.47±0.96 | 59.25±0.42 | **60.55±0.79** |
| H3K36me3 | 59.24±0.88 | 62.74±0.93 | 61.35±0.91 | 59.99±1.03 | 62.98±0.39 | 62.61±1.66 | **63.92±0.52** |
| H3K79me3 | 60.48±1.08 | 63.71±1.02 | 67.51±0.51 | 67.03±0.62 | **70.28±1.11** | 69.88±0.47 | 67.88±1.03 |
| H4 | 79.27±0.54 | 79.66±0.63 | 77.52±0.77 | 78.21±0.71 | 81.31±0.45 | 81.35±0.47 | **83.12±0.76** |
| H4ac | 46.17±2.59 | 49.52±1.62 | 56.46±1.11 | 57.09±1.68 | **62.10±0.99** | 59.54±0.86 | 56.91±0.39 |

### 4.1 Short Range Sequence Modeling

**Genomic Benchmark (Grešová et al., 2023)**
This benchmark involves a classification task on 8 datasets, we report the mean and standard deviation of accuracy over five runs with different random seeds. As shown in Table 1, our method outperforms the state-of-the-art on 7 of the 8 datasets, with particularly significant gains up to 5.75%.

Table 3: **Performance** $(\%, \uparrow)$ **comparison on Chromatin Profile Prediction.**

| Model | TF | DHS | HM |
|---|---|---|---|
| DNABERT-V2 (117M) | 96.16 | 92.50 | 86.18 |
| NT-V2 (500M) | 96.76 | 93.39 | 86.76 |
| HyenaDNA (6.5M) | 95.91 | 92.33 | 85.46 |
| ConvNova (27.4M) | 96.99 | 93.71 | 86.70 |
| HPB-DNA (6.5M) | **97.25±0.02** | **94.01±0.05** | **87.02±0.04** |

**Nucleotide Transformer Benchmark (Dalla-Torre et al., 2025)** It encompasses 18 datasets across histone marker prediction, regulatory annotation, and splice site annotation. For each, we report the mean and standard deviation over ten runs with different random seeds. The results in Table 2 highlight the effectiveness of our method. Focusing on models under 5M parameters, our approach achieves the best performance on 15 of the 18 tasks. When benchmarked against the entire field, including models exceeding 100M parameters, it still leads on 10 out of 18 tasks.

### 4.2 Long Range Sequence Modeling

**Chromatin Profile Prediction** This multi-label classification task aims to predict chromatin profiles and epigenetic markers from DNA sequences. We use the dataset from DeepSEA (Zhou & Troyanskaya, 2015), which includes 919 binary labels across transcription factor (TF) binding, DNase I-hypersensitive sites (DHS), and histone mark (HM) profiles. Following previous works (Nguyen et al., 2023; Bo et al., 2025), we measure performance using the AUROC score. As shown in Table 3, our model surpasses state-of-the-art methods in all profiles while having the smallest parameter count.

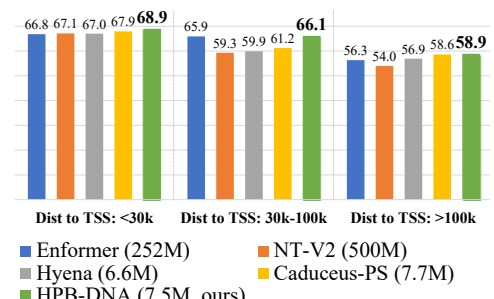

Figure 3: **Comparison on variant effects prediction.** AUROC $(\%, \uparrow)$ is reported for varying distances to the nearest Transcription Start Site (TSS): <30k, 30k-100k, >100k.

**Variant Effect Prediction** This task evaluates the model's ability to predict the impact of SNPs on gene expression over long sequences. Consistent with prior work (Schiff et al., 2024), all other methods use an input sequence length of 131k, while NT and Enformer use lengths of 12k and 196k, respectively. Features are extracted from the sequences using the model and then fed into an SVM for classification. The average results over five runs with different random seeds are reported. As shown in Fig. 3, our method outperforms competing approaches in all evaluated settings (specifically, 68.9±0.02, 66.1±0.01, and 58.9±0.01). This strong performance is achieved with a relatively small parameter count, highlighting the model's effectiveness in capturing long-range dependencies.

### 4.3 Ablation Study

**Plasticity Layers** We conduct ablations on LPL, DPL, and SPL separately to validate their contributions to the model. As shown in Table 4, removing any plasticity layer leads to a notable decline in performance. This confirms that the LPL, DPL, and SPL modules are structurally and functionally complementary; their synergistic integration is essential for achieving optimal results.

Table 4: **Ablation of each component of HPB.**

| | w/o LPL | w/o DPL | w/o SPL | HPB |
|---|---|---|---|---|
| Mouse Enhancers | 79.39±0.62 | 79.21±0.73 | 81.02±0.29 | **82.38±0.66** |
| Coding vs. Intergenomic | 91.06±0.38 | 89.98±0.55 | 92.11±0.16 | **94.46±0.47** |
| Human vs. Worm | 94.67±0.22 | 90.01±0.91 | 95.22±0.38 | **97.19±0.75** |
| Human Enhancers Cohn | 71.19±0.11 | 69.53±0.13 | 72.01±0.28 | **74.96±0.35** |
| Human Enhancers Ensembl | 88.66±0.62 | 85.91±0.51 | 90.29±0.08 | **91.08±0.61** |
| Human Regulatory | 90.34±0.55 | 89.19±0.49 | 91.36±0.15 | **94.20±0.23** |
| Human Nontata Promoters | 91.05±0.31 | 88.35±0.66 | 93.19±0.28 | **94.78±0.51** |
| Human OCR Ensembl | 78.61±0.76 | 76.20±0.85 | 79.38±0.49 | **81.95±1.01** |

**Local Feature Modeling** As shown in Table 5, LPL's advantage as a local feature extractor is evident from its performance against other strong models including CondConv (Yang et al., 2019) and GCB (Bo et al., 2025). This advantage stems from its token-specific feature extraction, a mechanism that better leverages local context to accurately capture diverse local patterns and ultimately improve model performance. For complete detailed results, please refer to Appendix A.11.

**Global Dependency Capturing** We compare the DPL with other modules capable of capturing global dependencies, such as long convolution and self-attention. Table 6 shows that the DPL achieves the best performance. We posit that this stronger result arises because multi-domain features provide a distinct and complementary source of information, enabling the model to capture dependencies that are imperceptible to purely single-domain operations. The complete detailed results are provided in Appendix A.12.

Table 5: **Performance** (%, ↑) **comparison of local motif extractors.**

| Model | CondConv | GCB | LPL |
|---|---|---|---|
| Mouse Enhancers | 79.89±0.41 | 80.38±0.68 | **82.38±0.66** |
| Human vs. Worm | 95.38±0.22 | 96.29±0.11 | **97.19±0.75** |
| Human Regulatory | 92.37±0.32 | 91.20±0.71 | **94.20±0.23** |

**Salient Feature Refining** We evaluate the efficacy of the SPL, SENet (Hu et al., 2018), and Hyena gating mechanisms from the **Information Bottleneck (IB)** (Tishby et al., 2000) perspective. As shown in Fig. 4, the x-axis represents the mutual information between the module's input and output features ($I(\text{input}; \text{output})$), where a lower value indicates greater information compression. The y-axis represents the mutual information between the output features and the true labels ($I(\text{output}; \text{label})$), where a higher value indicates more retained predictive information. We train the aforementioned models on the classification task for 200 epochs. The resulting training curves, plotted in Fig. 4, clearly show the benefits of SPL. Its trajectory simultaneously achieves the highest predictive power and the most significant information compression, demonstrating its efficacy as an efficient and effective feature refining mechanism.

Table 6: **Performance** (%, ↑) **comparison of 'long conv', 'self-attention' and 'DPL'.**

| Model | long conv | self attention | DPL |
|---|---|---|---|
| H3 | 79.86±0.38 | 78.55±0.51 | **82.03±0.38** |
| Promoter all | 95.73±0.53 | 96.02±0.22 | **97.01±0.71** |
| Splice Site All | 95.82±0.26 | 94.11±0.44 | **97.12±0.51** |

### 4.4 MODEL ANALYSIS

**Periodic Sequence Processing** We extract features from a nucleosome region DNA sequence (hg19, chr1:1005266-1007266), which is known to exhibit a dinucleotide periodicity of about 0.095 cycles/bp (a 10-10.5 bp period) (Reynolds et al., 2009). We then compute the Power Spectral Density (PSD) of the output features from different methods. As shown in Fig. 5(A), the HPB's features display a distinct dual peak at 0.0950 and 0.0955 cycles/bp, matching the specific periodicity. In contrast, the other models' outputs show no significant peak in this critical frequency band. This result highlights the sensitivity of our model to periodic signals in DNA, a direct benefit of its multi-domain modeling.

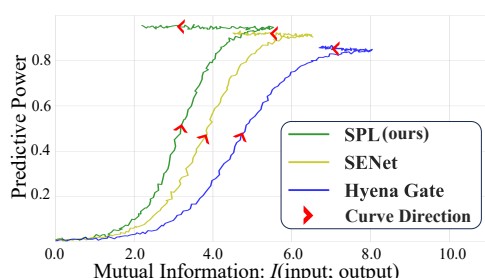

Figure 4: **Comparison of IB curves.** The curve direction shows the evolution of the x- and y-axis metrics during the training.

**Feature Interpretability Analysis** To probe the biological meaning of our model's pre-trained embeddings, we train a Sparse Autoencoder (SAE) (Bussmann et al., 2024) to disentangle them. We analyze the resulting sparse features for correlation with promoter regions (±2,000 bp around TSS). The analysis reveals multiple promoter-associated features, with feature channel 1990 being a prime example. As visualized for a specific locus in Fig. 5(B) (chr1:994976-1027744), the activation peaks of this feature show strong spatial alignment with the promoter regions, which confirms that our model autonomously identifies key regulatory elements like promoters from raw DNA sequences alone, without relying on explicit genomic annotations (see A.10 for more details).

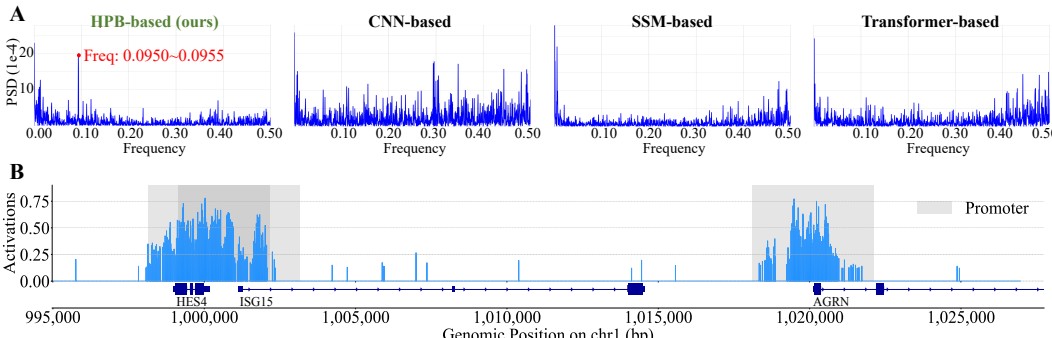

Figure 5: (A). **Power spectrum analysis** of periodic sequences processed by HPB-based and other models. (B). **Visualization of activations (blue) of a disentangled feature and promoter regions (gray).** Below are the IGV-style annotations for corresponding genes: HES4(-), ISG15(+), and ARGN(+). Arrows indicate transcription direction, with rectangles for CDS (thick) and UTR (thin).

## 5 Conclusion

In this work, we introduce the Holistic Protean Block (HPB), a novel architecture that flexibly models diverse patterns in DNA sequences through multi-level plasticity. Its Locus Plasticity Layer (LPL) captures fine-grained local features via token-specific convolutions. Its Domain Plasticity Layer (DPL) models global dependencies by fusing time- and frequency-domain features. Its Saliency Plasticity Layer (SPL) generates saliency scores for feature refinement at both the channel and position levels. By unifying these protean mechanisms, the HPB-based model (HPB-DNA) achieves state-of-the-art performance across comprehensive experiments, establishing a more powerful and principled paradigm for DNA sequence analysis.

**Limitations and Future Work**   The model and data scale in this work is modest due to resource constraints and for fair comparison. In the future, we will scale the model to billions of parameters on cross-species DNA and apply it to other domains like protein, RNA, and chemical sequences.

## 6 Ethics statement

This study is conducted using exclusively public data, and all genomic information is anonymized, posing no risk to individual privacy. Our proposed model is a computational tool developed to support and enhance the analysis of DNA sequences by scientific experts. Its scope is confined to research applications, and we have determined that it does not present foreseeable risks of adverse social or ethical outcomes. We are committed to the responsible dissemination and application of our research within the scientific community.

## 7 Reproducibility statement

To ensure the reproducibility of the results presented in this paper, we have provided a detailed description of our methodology in the 'METHODOLOGY' section and a comprehensive overview of the datasets and experimental settings in the 'EXPERIMENTS' section. Furthermore, we provide extensive implementation details in the 'APPENDIX' section. These include, but are not limited to: (1) the specific architectural details for constructing the HPB-DNA model; (2) the data preprocessing steps for each benchmark dataset; and (3) a complete list of training hyperparameters for all experiments. We believe these details are sufficient to allow for a full and straightforward replication of our findings. To further facilitate reproducibility and encourage future research, we commit to making our source code, pre-trained models, and data processing scripts publicly available upon acceptance of this paper.

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

# A  APPENDIX

## A.1  HPB-BASED DNA MODELS

As shown in Fig. 6, in the 'Pre-Norm' architecture, layer normalization is applied before each sub-layer. The input is first normalized to **E** and then processed by the HPB. After the first residual connection, the result is normalized again before being passed to the FFN, followed by a final residual connection. In the 'Post-Norm' architecture, layer normalization is applied after each residual connection. The input feature map **E** first passes through the HPB module. The output is then added to the input via a residual connection, and the result is normalized. This process is repeated for the subsequent FFN.

Figure 6: Architectural scaling of the HPB to create DNA models of different capacities.

## A.2  DETAILS OF PRE-TRAINING

During the pre-training phase, we trained the model for 10 epochs on the human genome (hg38) using the Adam optimizer, a batch size of 256, and a learning rate of 1e-4. We used a 1-mer tokenization ($k = 1$), where each nucleotide was treated as a single token. Within the LPL, we used convolutional kernel bases with sizes of [1, 3, 5, 7]. In the DPL, the wavelet transform was configured with 3 decomposition levels. In the SPL, the kernel size of the depthwise separable convolution was $7 \times 7$. All experiments were conducted on 8 NVIDIA A800 GPUs.

## A.3  GENOMIC BENCHMARK

For this benchmark, we constructed a DNA model using an encoder composed of 2 HPBs with a feature dimension of 116, stacked in a 'pre-norm' configuration. Consistent with prior work, the model was pre-trained on the human reference genome (hg38) using a sequence length of 1K. During fine-tuning, we used a batch size of 128 and a learning rate of 6e-4 for each task. For methods with multiple architectural variants (e.g., Caduceus and xLSTM), we selected their best-performing architecture for the specific task in our comparison.

## A.4 Nucleotide Transformer Benchmark

For this benchmark, we constructed a DNA model with an encoder composed of 4 HPBs and a feature dimension of 168. Consistent with prior work, the model was pre-trained on the human reference genome (hg38) using a sequence length of 1K. During the fine-tuning phase, we used a batch size ranging from 128 to 256 and a learning rate between 1e-4 and 1e-3. The detailed hyperparameters for each task are summarized in Table 7. For methods with multiple architectural variants (e.g., Caduceus and xLSTM), we selected their best-performing architecture for the specific task in our comparison.

Table 7: **Hyperparameter settings for Nucleotide Transformer benchmark.**

|  |  | **Learning Rate** | **Batch Size** |
|---|---|---|---|
| Regulatory | Promoter all | 2e-4 | 256 |
|  | Promoter non-TATA | 2e-4 | 256 |
|  | Promoter TATA | 2e-4 | 256 |
|  | Enhancer | 6e-4 | 128 |
|  | Enhancer types | 6e-4 | 128 |
| Splice Site | Splice acceptor | 6e-4 | 128 |
|  | splice donor | 6e-4 | 128 |
|  | splice all | 6e-4 | 128 |
| Histone | H3 | 1e-4 | 256 |
|  | H3K4me1 | 1e-3 | 128 |
|  | H3K4me2 | 1e-3 | 128 |
|  | H3K4me3 | 1e-3 | 128 |
|  | H3K9ac | 8e-4 | 128 |
|  | H3K14ac | 8e-4 | 128 |
|  | H3K36me3 | 1e-3 | 128 |
|  | H3K79me3 | 6e-4 | 256 |
|  | H4 | 1e-4 | 256 |
|  | H4ac | 6e-4 | 256 |

## A.5 Convolutional Kernel Bases

We conducted an ablation study on the Genomic Benchmark to assess the impact of varying the number of convolutional kernel bases, with the results detailed in Table 8. The findings indicate that performance improves with more kernel bases, but the gains diminish when four or more are used. This suggests that, for this dataset, four kernel bases of different sizes are adequate for capturing a comprehensive set of meta-features, leading to the construction of features with high discriminative capabilities.

Table 8: **Performance (%, ↑) comparison of different sizes of 'conv kernel bases'.**

| Conv Kernel Bases | [1] | [1, 3, 5] | [1, 2, 3] | [1, 3, 5, 7] | [1, 3, 4, 5, 7] |
|---|---|---|---|---|---|
| Mouse Enhancers | 80.01±0.63 | 81.93±0.49 | 81.12±0.52 | 82.38±0.66 | 81.92±0.77 |
| Human vs. Worm | 96.89±0.71 | 96.96±0.76 | 96.91±0.69 | 97.19±0.75 | 97.25±0.81 |
| Human Regulatory | 93.88±0.28 | 94.16±0.31 | 94.01±0.26 | 94.20±0.23 | 94.15±0.33 |
| Coding vs. Intergenomic | 93.86±0.41 | 94.35±0.47 | 94.43±0.58 | 94.46±0.47 | 94.59±0.71 |
| Human Enhancers Cohn | 74.93±0.32 | 74.96±0.29 | 74.81±0.41 | 74.96±0.35 | 74.85±0.27 |
| Human Enhancers Ensembl | 90.28±0.58 | 90.96±0.49 | 90.83±0.62 | 91.08±0.61 | 90.91±0.73 |
| Human Nontata Promoters | 93.12±0.46 | 94.39±0.53 | 94.82±0.61 | 94.78±0.51 | 94.10±0.50 |
| Human OCR Ensembl | 80.56±0.86 | 81.88±0.79 | 81.39±0.89 | 81.95±1.01 | 81.27±0.92 |

## A.6 Wavelet levels

We evaluated the model's performance across different numbers of wavelet decomposition levels, with the results presented in Table 9. The results indicate that, on the whole, performance improves with more levels, a benefit derived from the increasingly fine-grained frequency resolution. Nevertheless, these gains become marginal when the number of levels exceeds three. Therefore, to strike a balance between performance and computational efficiency, we set the number of wavelet decomposition levels to $J=3$ for our experiments.

Table 9: **Performance (%, ↑) comparison of different wavelet levels $J$.**

| Wavelet Levels $J$ | 1 | 2 | 3 | 4 | 5 |
|---|---|---|---|---|---|
| Mouse Enhancers | 81.11±0.64 | 81.66±0.69 | 82.38±0.66 | 81.91±0.78 | 81.95±0.53 |
| Human vs. Worm | 97.01±0.39 | 97.03±0.73 | 97.19±0.75 | 97.05±0.75 | 97.03±0.77 |
| Human Regulatory | 93.88±0.31 | 93.91±0.26 | 94.20±0.23 | 94.12±0.25 | 94.19±0.39 |
| Coding vs. Intergenomic | 94.41±0.47 | 94.39±0.44 | 94.46±0.47 | 94.46±0.51 | 94.46±0.46 |
| Human Enhancers Cohn | 74.81±0.32 | 74.93±0.41 | 74.96±0.35 | 74.95±0.38 | 74.93±0.29 |
| Human Enhancers Ensembl | 90.88±0.57 | 91.02±0.60 | 91.08±0.61 | 91.07±0.63 | 91.12±0.72 |
| Human Nontata Promoters | 93.99±0.55 | 94.26±0.83 | 94.78±0.51 | 94.79±0.78 | 94.76±0.58 |
| Human OCR Ensembl | 81.90±0.89 | 81.93±0.92 | 81.95±1.01 | 81.95±1.08 | 81.93±0.97 |

## A.7 Chromatin Profile Prediction

The prediction of epigenetic markers in non-coding regions is a fundamental task for elucidating the functional impact of disease-associated variants. For this work, we utilized the benchmark dataset established by DeepSEA (Zhou & Troyanskaya, 2015). We followed the official data partitioning scheme, which ensures a strict, non-overlapping split between training and testing sets based on chromosomes. The dataset comprises 919 chromatin features compiled from the ENCODE (Consortium et al., 2012) and Roadmap (Kundaje et al., 2015), including 690 transcription factor binding profiles (TF), 125 DNase I-hypersensitive sites (DHS), and 104 histone mark (HM) profiles. Each sample consists of a 1000 bp DNA sequence. A binary label of 1 was assigned for a given feature if the central 200-bp region of the sequence overlapped by more than 50% with a corresponding peak region; otherwise, the label was 0. This process generated a 919-dimensional label vector for each sample, framing the problem as a large-scale multi-label classification task. As the original data were based on the hg19 reference genome, we used the LiftOver tool (Kent et al., 2002) to convert all genomic coordinates to hg38. Samples that failed the conversion or resulted in an inconsistent sequence length (approximately 0.5%) were discarded.

For this task, we constructed a DNA model using an encoder of 6 HPB layers with a feature dimension of 256. The model was trained with a batch size of 512 and a learning rate of 4e-4.

## A.8 Variant Effect Prediction

Our methodology for predicting the causal impact of SNPs on gene expression followed the established protocol from prior work (Schiff et al., 2024) and involved four key stages. First, SNPs with a SuSiE-derived (Wang et al., 2020a) causal probability greater than 0.9 (Avsec et al., 2021) were assigned a positive label for the task. Second, we structured the dataset by using chromosomes 9 and 10 as a held-out test set (Kao et al., 2024) and stratifying all data based on the SNP's distance to the nearest Transcription Start Site (TSS). Third, for feature extraction, we fed each SNP's full-length reference and alternative sequences (12k bp for Nucleotide Transformer, 196k bp for Enformer, and 131k bp for the other methods) into the accordingly pre-trained model (a 6-layer HPB encoder with a 256-dimensional feature space) to generate embeddings. These were then averaged within a 1536 bp window centered on both the reference and alternative SNP sequences, and concatenated with the tissue of origin. Finally, we trained a Support Vector Machine (SVM) with an RBF kernel on 5,000 random data points from each TSS distance stratum, evaluated performance using AUROC on the test set. The average results are reported over five independent runs. For methods with multiple architectural variants (e.g., Caduceus), we selected their best-performing architecture for the specific task in our comparison.

A.9 DETAILS OF PERIODIC SEQUENCE PROCESSING

We performed feature extraction and Power Spectral Density (PSD) analysis on nucleosome-bound DNA sequences using ConvNova, Caduceus, and DNABERT-V2, which correspond to the CNN-based, Mamba-based, and Transformer-based methods in Fig. 5, respectively.

A.10 DETAILS OF FEATURE INTERPRETATION

As mentioned previously, we sought to decompose HPB's pre-trained dense embeddings into more disentangled, meaningful biological features. To this end, we trained a BatchTopK Sparse Autoencoder (SAE) (Bussmann et al., 2024) to learn a disentangled sparse representation from our model's dense embeddings. The SAE was configured with a dictionary size of 8192 features and trained with an L1 regularization coefficient of 3e-4 to enforce sparsity. The training data for the SAE consisted of hidden state embeddings of 1 billion tokens extracted from the final layer of the HPB model.

We then systematically scanned the entire genome to identify which of the resulting SAE features were specifically associated with promoter regions. Here we defined promoter regions as the ±2,000 bp window around each Transcription Start Site (TSS) based on the GENCODE v48 annotation (Frankish et al., 2020). We employed a sliding window strategy with a stride of 8,192 bp to process the genome and fed the corresponding DNA sequence into the pre-trained HPB model to obtain its embedding, which was then passed through the trained SAE to compute the activation value for each of the 8192 features. This process yielded a genome-wide activation map for each feature. We specifically hypothesized that some features would be associated with promoter regions. To score the association for each feature, we computed the log-ratio of its activation values within promoter regions compared to non-promoter regions:

$$S_f = \log \left( \frac{\mathrm{mean}(A_f^{\mathrm{promoter}})}{\mathrm{mean}(A_f^{\mathrm{non\text{-}promoter}}) + \epsilon} \right) \tag{6}$$

where $A_f^{\mathrm{promoter}}$ and $A_f^{\mathrm{non\text{-}promoter}}$ are the collections of activation values for feature $f$ in promoter and non-promoter regions, respectively. A small constant $\epsilon$ was added for numerical stability. A high positive score $S_f$ indicates that the feature $f$ is a strong marker for promoters.

A.11 LOCAL MOTIF EXTRACTORS

As shown in Table 10, LPL maintains a clear performance advantage over these state-of-the-art local feature extractors across all the datasets in Genomic Benchmark.

Table 10: **Performance (%, ↑) comparison of local motif extractors.**

| Model | Conv | CondConv (Yang et al., 2019) | ConvNeXt | MambaOut (Yu & Wang, 2025) | GCB (Bo et al., 2025) | LPL |
|---|---|---|---|---|---|---|
| Mouse Enhancers | 80.01±0.19 | 79.89±0.41 | 80.22±0.22 | 78.06±0.39 | 80.38±0.68 | **82.38±0.66** |
| Coding vs. Intergenomic | 92.33±0.09 | 92.11±0.18 | 92.38±0.31 | 93.19±0.23 | 93.23±0.14 | **94.46±0.47** |
| Human vs. Worm | 96.36±0.10 | 95.38±0.22 | 96.12±0.08 | 95.87±0.26 | 96.29±0.11 | **97.19±0.75** |
| Human Enhancers Cohn | 73.36±0.15 | 73.39±0.27 | 72.58±0.61 | 73.01±0.25 | 73.03±0.37 | **74.96±0.35** |
| Human Enhancers Ensembl | 89.78±0.06 | 89.66±0.12 | 87.85±0.44 | 88.03±0.38 | 87.29±0.25 | **91.08±0.61** |
| Human Regulatory | 91.12±0.27 | 92.37±0.32 | 91.99±0.39 | 92.96±0.78 | 91.20±0.71 | **94.20±0.23** |
| Human Nontata Promoters | 94.25±0.04 | 94.12±0.11 | 94.20±0.13 | 94.13±0.21 | 94.32±0.42 | **94.78±0.51** |
| Human OCR Ensembl | 80.01±0.26 | 79.95±0.31 | 80.81±0.21 | 80.35±0.26 | 80.11±0.28 | **81.95±1.01** |

A.12 GLOBAL DEPENDENCY CAPTURING

As shown in the results in Table 11, our proposed DPL consistently outperforms both Long Convolution (the core module of Hyena) and Self-Attention (the core mechanism of Transformers) across this entire benchmark.

A.13 SCALABILITY

To further demonstrate the scalability of our architecture, we scaled our model to 100M and 500M (0.5B) parameters and evaluated them on the NT dataset. As shown in the table below, the results

Table 11: **Performance ($\%, \uparrow$) comparison of 'long conv', 'self-attention' and 'DPL'.**

| Model | long conv | self attention | DPL |
|---|---|---|---|
| Promoter all | 95.73±0.53 | 96.02±0.22 | **97.01±0.71** |
| Non-TATA | 95.12±0.29 | 95.23±0.31 | **96.88±0.16** |
| TATA | 96.03±0.11 | 96.28±0.40 | **96.59±0.42** |
| Enhancer | 53.10±0.25 | 52.98±0.49 | **58.69±0.49** |
| Enhancer types | 48.66±0.59 | 47.98±0.31 | **50.87±0.22** |
| Splice Site Acceptor | 95.99±0.12 | 96.36±0.08 | **97.16±0.11** |
| Splice Site Donor | 96.39±0.14 | 96.88±0.21 | **97.79±0.25** |
| Splice Site All | 95.82±0.26 | 94.11±0.44 | **97.12±0.51** |
| H3 | 79.86±0.38 | 78.55±0.51 | **82.03±0.38** |
| H3K4me1 | 54.02±0.13 | 53.11±0.55 | **55.39±0.56** |
| H3K4me2 | 51.93±0.27 | 53.76±0.19 | **53.89±0.23** |
| H3K4me3 | 50.32±0.35 | 49.39±0.41 | **54.86±0.68** |
| H3K9ac | 60.89±0.99 | 59.45±0.86 | **64.50±1.19** |
| H3K14ac | 56.91±0.59 | 57.87±0.66 | **60.55±0.79** |
| H3K36me3 | 62.08±0.11 | 62.66±0.23 | **63.92±0.52** |
| H3K79me3 | 65.31±0.38 | 66.85±0.12 | **67.88±1.03** |
| H4 | 78.27±0.57 | 79.33±0.61 | **83.12±0.76** |
| H4ac | 56.56±0.15 | 55.99±0.05 | **56.91±0.39** |

Table 12: Performance comparison on the Nucleotide Transformer benchmark. We compare our HPB-DNA models (scaled from 1.9M to 500M) against the NT-V2 (500M).

| Model | NT-V2 (500M) | HPB-DNA (1.9M) | HPB-DNA (100M) | HPB-DNA (500M) |
|---|---|---|---|---|
| Promoter all | 97.61±0.12 | 97.01±0.71 | 97.85±0.25 | **98.23±0.19** |
| Non-TATA | 97.53±0.14 | 96.88±0.16 | 97.98±0.09 | **98.56±0.03** |
| TATA | 96.35±1.06 | 96.59±0.42 | 98.02±0.19 | **99.15±0.09** |
| Enhancer | 56.62±3.20 | 58.69±0.49 | 59.96±0.31 | **61.29±0.22** |
| Enhancer types | 45.95±2.30 | 50.87±0.22 | 52.88±0.21 | **62.97±0.15** |
| Splice Site Acceptor | 98.64±0.22 | 97.16±0.11 | 98.18±0.12 | **99.06±0.03** |
| Splice Site Donor | 98.86±0.18 | 97.79±0.25 | 98.76±0.21 | **99.28±0.14** |
| Splice Site All | 98.37±0.14 | 97.12±0.51 | 98.57±0.39 | **99.61±0.23** |
| H3 | 79.14±1.21 | 82.03±0.38 | 84.01±0.19 | **85.22±0.08** |
| H3K4me1 | 54.21±0.74 | 55.39±0.56 | 57.09±0.31 | **59.21±0.16** |
| H3K4me2 | 32.12±2.99 | 53.89±0.23 | 57.38±0.21 | **60.06±0.12** |
| H3K4me3 | 40.25±3.09 | 54.86±0.68 | 60.88±0.46 | **66.92±0.28** |
| H3K9ac | 56.62±0.88 | 64.50±1.19 | 66.29±0.68 | **68.13±0.39** |
| H3K14ac | 54.07±1.67 | 60.55±0.79 | 65.38±0.55 | **71.29±0.31** |
| H3K36me3 | 62.74±0.93 | 63.92±0.52 | 66.61±0.41 | **69.32±0.22** |
| H3K79me3 | 63.71±1.02 | 67.88±1.03 | 69.55±0.53 | **72.98±0.16** |
| H4 | 79.66±0.63 | 83.12±0.76 | 84.91±0.62 | **85.85±0.19** |
| H4ac | 49.52±1.62 | 56.91±0.39 | 61.29±0.77 | **66.69±0.71** |

reveal a clear scaling law: the 100M model achieves substantial performance gains across all tasks compared to the 1.9M variant, and the 500M model yields further notable improvements over the 100M model. Most notably, our 500M model outperforms NT-v2 at a comparable parameter scale. This finding strongly validates that our proposed architecture is not only highly effective but also scales more favorably than alternative genomic foundation models.

## A.14 MODEL EFFICIENCY

We conduct a comprehensive empirical evaluation of FLOPs, memory footprint, and runtime (Table 13). We performed a fair comparison against Mamba, Hyena, a standard Transformer, and a Transformer with Flash Attention. All models were configured with approximately 12M parameters and evaluated on input sequences of 100k length. Training and inference tests for single samples were conducted on a single Nvidia A800 GPU.

As shown in the results below, our model's FLOPs are significantly lower than the Transformer's and are even slightly lower than those of Hyena and Mamba. This empirically validates the sub-linear computational complexity of our approach. Regarding memory usage and wall-clock time,

Table 13: **Efficiency Comparison.** (a) Training (Forward + Backward) and (b) Inference (Forward Only) on sequences of length 100k.

| Model | FLOPs | VRAM | Wall Clock |
|---|---|---|---|
| *(a) Training - Forward + Backward* | | | |
| Mamba | 5,619 G | 5.8 GB | 0.46s |
| Hyena | 5,573 G | 6.2 GB | 0.89s |
| Standard Transformer | 517,328 G | *OOM* | N/A |
| Transformer + FlashAttn | 516,931 G | 7.8 GB | 4.87s |
| HPB (Ours) | 5,431 G | 6.8 GB | 0.93s |
| *(b) Inference - Forward Only* | | | |
| Mamba | 1,452 G | 0.5 GB | 0.13s |
| Hyena | 1,391 G | 1.1 GB | 0.26s |
| Standard Transformer | 125,392 G | 28.7 GB | 19.30s |
| Transformer + FlashAttn | 125,281 G | 2.4 GB | 1.30s |
| HPB (Ours) | 1,316 G | 1.2 GB | 0.29s |

Table 14: Performance comparison on the Gene Finding task (Bend dataset). Metric: Matthews Correlation Coefficient (MCC).

| Model | NT-H (500M) | DNABERT-2 (117M) | GENA-LM (336M) | HyenaDNA (6.5M) | ConvNova (7.4M) | HPB-DNA (6.6M) |
|---|---|---|---|---|---|---|
| MCC | 0.41 | 0.43 | 0.52 | 0.35 | 0.55 | **0.68±0.03** |

our model is more efficient than the Transformer, though currently slightly higher than Mamba and Hyena. It is crucial to note that architectures like Transformer (Flash Attention) benefit from highly mature, engineering-level accelerations. Our current implementation has not yet undergone such low-level optimization. Given our lower FLOPs, this indicates substantial potential for further acceleration, which we prioritize as future work.

In summary, our model surpasses the Transformer in efficiency and remains competitive with Mamba and Hyena. Considering our model simultaneously achieves better accuracy on both short- and long-sequence tasks, we believe it represents a highly promising and efficient architectural alternative.

## A.15 BEND GENE FINDING

To evaluate the model's capacity for complex genomic annotation, we conduct experiments on Gene finding (Marin et al., 2023). This task requires classifying nucleotides into functional roles (e.g., exons, introns, donors, acceptors) based on the GENCODE dataset (Harrow et al., 2012), which features sequences up to 14,000 bps in length. This presents a dual challenge: models must detect fine-grained local signals to identify exon-intron boundaries while also modeling long-range dependencies to maintain global structural consistency. The results, presented in Table 14, demonstrate that our model holds a clear and significant advantage over other state-of-the-art methods on this new benchmark as well.

## A.16 FEATURE ALIGNMENT

To evaluate the feature quality learned by different models, we pretrained our HPB-DNA model (12.3M) alongside a Transformer (12.6M) and a Hyena model (12.8M), using hg38 DNA sequences with a unified 32k context length. For each model, we trained a corresponding BatchTopK SAE (Bussmann et al., 2024) with consistent hyperparameters to disentangle its representations and evaluated their alignment with genomic annotations (promoters, exons, and CDS regions). A genomic position was considered "activated" if the feature value was non-zero. To quantify performance, we calculated the F1 score for each feature and reported the average F1 score of the top-5 most relevant features for each genomic region type. As shown in Table 15, under this strictly controlled setting, our model captures genomic regulatory information more effectively than the compared methods. Furthermore, we have also provided the alignment of the features output by HPB-DNA with the CDS regions of the DNA sequence for reference, Fig. 7.

Table 15: **F1 Score between Features and Genomic Regions.**

|  | Transformer | Hyena | HPB |
|---|---|---|---|
| promoter | 0.0601 | 0.0900 | **0.1320** |
| exons | 0.0758 | 0.0830 | **0.0984** |
| cds | 0.0341 | 0.0500 | **0.0719** |

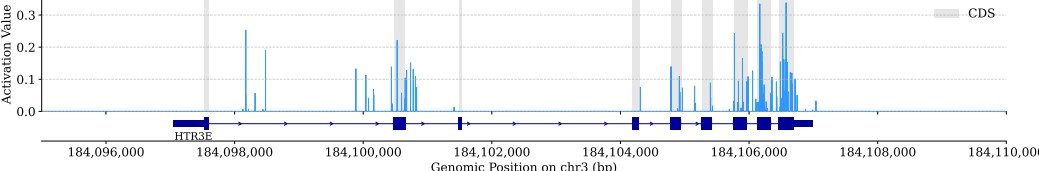

Figure 7: Visualization of feature alignment with cds regions.

### A.17 SEQUENCE LENGTH OF PRE-TRAINING

We pre-trained our model on the hg38 DNA sequence using two different sequence lengths: 1K and 12K (while maintaining the same model size). We then evaluated both pre-trained models on the Genomic Benchmark and the Nucleotide Transformer Benchmark. The results are summarized below (Table 16 and Table 17).

The results clearly indicate that our model pre-trained on 12K sequences generally outperforms the model pre-trained on 1K sequences, either exceeding or matching the performance across multiple datasets.

We hypothesize that this improvement stems from the fact that pre-training on shorter sequences necessarily truncates long-range dependency information, which prevents the model from learning complete contextual associations. Utilizing a longer sequence length alleviates this issue, allowing the model to learn more accurate and holistic context-aware information. This outcome further validates that the "protean" nature of our architecture extends beyond mere feature pattern variability to encompass robustness and generalization across sequence lengths.

Table 16: Ablation of pre-training length on Genomic Benchmark.

| Model | HPB-DNA (1K length) | HPB-DNA (12K length) |
|---|---|---|
| Mouse Enhancers | 82.38±0.66 | 84.15±0.71 |
| Coding vs. Intergenomic | 94.46±0.47 | 94.49±0.42 |
| Human vs. Worm | 97.19±0.75 | 97.16±0.68 |
| Human Enhancers Cohn | 74.96±0.35 | 75.23±0.29 |
| Human Enhancers Ensembl | 91.08±0.61 | 91.68±0.65 |
| Human Regulatory | 94.20±0.23 | 94.26±0.31 |
| Human Nontata Promoters | 94.78±0.51 | 94.98±0.16 |
| Human OCR Ensembl | 81.95±1.01 | 83.01±0.95 |

### A.18 PERIODIC SIGNAL INTERFERENCE

To validate whether the multi-domain mechanism will introduce noise, we constructed contrastive groups (with vs. without periodic signals) derived from the Human Enhancers Cohn and Human Nontata Promoters datasets (Genomic Benchmark). In the groups containing periodic signals, we appended a 500-token random periodic signal (period = 5bp) to the end of each sequence sample. In the groups without periodic signals, we appended 500 non-periodic random bases. We then trained and evaluated the model on these datasets, where the objective remained enhancer or promoter prediction (independent of the appended signals). We evaluated both classification accuracy and

Table 17: Ablation of pre-training length on Nucleotide Transformer Benchmark.

| Model | HPB-DNA (1K length) | HPB-DNA (12K length) |
|---|---|---|
| Promoter all | 97.01±0.71 | 97.65±0.28 |
| Non-TATA | 96.88±0.16 | 97.16±0.09 |
| TATA | 96.59±0.42 | 97.08±0.45 |
| Enhancer | 58.69±0.49 | 59.31±0.52 |
| Enhancer types | 50.87±0.22 | 52.47±0.16 |
| Splice Acceptor | 97.16±0.11 | 97.52±0.15 |
| Splice Donor | 97.79±0.25 | 97.76±0.06 |
| Splice All | 97.12±0.51 | 97.68±0.31 |
| H3 | 82.03±0.38 | 83.11±0.12 |
| H3K4me1 | 55.39±0.56 | 56.19±0.51 |
| H3K4me2 | 53.89±0.23 | 53.95±0.21 |
| H3K4me3 | 54.86±0.68 | 55.09±0.57 |
| H3K9ac | 64.50±1.19 | 64.82±0.92 |
| H3K14ac | 60.55±0.79 | 61.33±0.66 |
| H3K36me3 | 63.92±0.52 | 64.28±0.58 |
| H3K79me3 | 67.88±1.03 | 68.19±0.87 |
| H4 | 83.12±0.76 | 83.36±0.55 |
| H4ac | 56.91±0.39 | 57.87±0.49 |

the feature response ratio (defined as the ratio of feature activation intensity in the target region relative to the padding region), reporting the mean results over 5 independent runs. As shown in Table 18, the test accuracies on both groups are highly comparable. Crucially, for the group with periodic signals, the feature response ratio is extremely high and aligns closely with that of the group without periodic signals. These results demonstrate that, for sequences either with or without periodic signals, our model effectively suppresses regions irrelevant to the prediction target rather than introducing multi-domain noise.

Table 18: Comparison of sequences with or without periodic signal interference on `Human Enhancers Cohn` and `Human Nontata Promoters`.

| Dataset | Condition | Accuracy (%) | Feature Response Ratio |
|---|---|---|---|
| `Human Enhancers Cohn` | With periodic signal | 75.36±0.21 | 1218.03 |
| | Without periodic signal | 75.29±0.19 | 1206.01 |
| `Human Nontata Promoters` | With periodic signal | 95.58±0.08 | 679.38 |
| | Without periodic signal | 95.62±0.06 | 682.59 |

## A.19   LLM USAGE

An LLM is only used for grammatical refinement. The authors take full responsibility for this paper.

