# OpenReview forum: "Holistic protean block for long-range DNA sequence modeling"
_ICLR.cc/2026/Conference — Submitted to ICLR 2026_

### Official Review · Reviewer_oS3Z · 2025-10-24

**Soundness:** 3
**Presentation:** 2
**Contribution:** 2
**Rating:** 4
**Confidence:** 3

**Summary:**

The authors introduced a Holistic Protean block (HPB) as a new building block for long context gLMs. HPB combines three modules including the Locus Plasticity Layer that learns token-specific convolution filters, the Domain Plasticity Layer (DPL) that fuses local and long convolutions, and the saliency plasticity layer (SPL) that performs dual-axis feature gating mechanism. By stacking these three types of blocks, HPB aims to capture both local and long range interactions of long genomics sequences while having quasi-linear time complexity. The proposed model is tested on Genomics Benchmarks and NT benchmark. Author also show improvements on chromatin profile prediction and a long-context VEP task.

**Strengths:**

1. HPB's design unifies two convolution block variants and a gated mechanism into a single operator block. This enables the model to capture both short and long-range interaction of the long sequence. Benchmarking on both short and long range tasks show improvements comparing to existing DNA LMs.

2. The model demonstrate also improvement on a VEP prediction task, surpassing both existing DNA lms and a supervised expert model of Enformer.

**Weaknesses:**

1. Despite the holistic framing, each HPB component echoes a previously known design. The LPL is essentially a Conditional Conv layer, and DPL is pretty similar to Hyena's long-convolution operator. The motivation for combining these components in this way is questionable.

2. The improvements on the benchmarks seems minimal and doesn't show significance in statistics. Maybe harder benchmarks like DNALongBench/BEND can be considered to better show benchmark result.

3. Though the paper states about quasi-linear computation time, it doesn't show an efficiency analysis section comparing to other architectures.

**Questions:**

1. Can the author evaluates their model on some newer and harder benchmarks like BEND or DNALongBench?

2. Can the author add an analysis section about training/inference throughput and memory consumption to show the performance potential of this HPB architecture comparing to the other existing ones like Mamba, Hyena or Transformers with flash attention?

---

> ### Author Response · Authors · 2025-11-21
> **Part 1 of the responses**
>
> ***We sincerely appreciate your valuable review comments and the effort you have put into them. We have addressed each of the questions you raised in detail. If anything is unclear, please inform us, and we will respond promptly.***
> >W1. Motivation and contribution of each component.
>
> ***Response:*** We elaborate on two key aspects: how our module design fundamentally differs from existing methods and the motivation behind them:
> *For LPL:* The differences between LPL and Conditional Convolution (CondConv) are twofold:
> 1. CondConv [1] is **sample-specific**, generating one kernel per sample, every token in that sample is processed by the same dynamically-generated kernel. LPL is **token-specific**, generating unique mixing weights per token. This finer-grained adaptability is essential for handling the diverse local patterns in a single DNA sequence (as shown in A.11 of the revised paper).
> 2. CondConv combines parameters (kernel weights) to create a single dynamic kernel. LPL combines features (the "meta-features") that result from multiple, fixed kernel bases. This feature-space combination allows LPL to dynamically fuse multi-scale information at every position, a capability CondConv lacks.
>
> We believe this token-specific, multi-scale feature-fusion mechanism is a novel contribution to genomic sequence analysis.
>
> *For DPL:* The distinction between DPL and Hyena's long-convolution [2] lies in its handling of signal domains. Long-convolution operates strictly in the time domain and is therefore less sensitive to frequency-specific signals like nucleosome periodicity (as stated in Lines 76-80 of the initial submission). Conversely, DPL extracts features from **both time and frequency domains** and adaptively fuses them. This enables the simultaneous capture of time-sensitive and frequency-sensitive information, a critical capability for DNA analysis that Hyena lacks. This is evidenced by the frequency analysis in Figure 5(A) and the stronger performance of DPL in A.12.
>
> Ultimately, the core novelty of our work is the synergistic integration of LPL, DPL, and SPL into a unified "multi-level plasticity" framework. This design addresses the specific complexity of DNA, which consists of local motifs, long-range dependencies, and periodic patterns (as stated in Lines 11-12 and Lines 107-110 of the initial submission). Since no single existing operator can capture all these features simultaneously, our holistic approach provides a necessary solution.
>
> *references*
> [1] CondConv: Conditionally Parameterized Convolutions for Efficient Inference, NeurIPS 2019.
> [2] HyenaDNA: Long-Range Genomic Sequence Modeling at Single Nucleotide Resolution, NeurIPS 2023.
>
> >W2&Q1. Evaluation on harder benchmarks.
>
> ***Response:*** First, we would like to reiterate the notable advantages our model has already demonstrated: it achieves **up to a 5.75%** improvement over SOTA methods on the Genomic Benchmark (Human Regulatory), a 1.81% gain on the challenging Nucleotide Transformer Benchmark (H4), and **a lead under all settings of ultra-long sequence tasks like Variant Effect Prediction**.
> To further validate our method's superiority, we have, as you suggested, conducted additional comparative experiments on the Bend benchmark (please see A.15 for more details). The results (shown below) indicate that our method **outperforms the current SOTA by 13%**. This substantial advantage provides further, robust evidence of our method's effectiveness in processing long-range sequences. We have included this result in the A.15 of the revised paper.
>
> Performance comparison on Gene Finding benchmark (BEND):
> | | NT-H (500M) | DNABERT-2 (117M) | GENA-LM (336M) | HyenaDNA (6.5M) |ConvNova (7.4M) |HPB-DNA (6.6M) |
> | :--- | :--- | :--- | :--- | :--- |:--- |:--- |
> | MCC | 0.41 | 0.43 | 0.52 | 0.35 | 0.55 | **0.68** |

---

> ### Author Response · Authors · 2025-11-21
> **Part 2 of the responses**
>
> >W3&Q2. Model efficiency of HPB.
>
> ***Response:*** While our initial submission provided a theoretical analysis of computational complexity (Lines 298-302), we have now followed your recommendation to conduct a comprehensive empirical evaluation of FLOPs, memory footprint, and runtime.
> We performed a fair comparison against Mamba, Hyena, a standard Transformer, and a Transformer with Flash Attention. All models were configured with 12M parameters and evaluated on input sequences of 100k length. Training and inference tests for a single sample were conducted on the Nvidia A800 GPU.
> As shown in the results below, our model's FLOPs are **significantly lower** than the Transformer's and are even slightly lower than those of Hyena and Mamba. This empirically validates the quasi-linear computational complexity of our approach.
> Regarding memory usage and wall-clock time, our model is more efficient than the Transformer, though currently slightly higher than Mamba and Hyena. It is crucial to note that architectures like Transformer (Flash Attention) benefit from highly mature, engineering-level accelerations. Our current implementation has not yet undergone such low-level optimization. Given our lower FLOPs, this indicates substantial potential for further acceleration, which we prioritize as future work.
> In summary, our model surpasses the Transformer in efficiency and remains competitive with Mamba and Hyena. Considering our model simultaneously achieves better accuracy on both short- and long-sequence tasks, we believe it represents a highly promising and efficient architectural alternative. These results have been included in the revised paper (please see A.14).
>
> *Training - Forward + Backward:*
> | Model | FLOPs | VRAM | Wall Clock |
> | :--- | :--- | :--- | :--- |
> | Mamba | 5,619 G | 5.8 GB | 0.46s |
> | Hyena | 5,573 G | 6.2 GB | 0.89s |
> | Standard Transformer| 517,328 G | OOM | N/A |
> | Transformer + FlashAttn | 516,931 G | 7.8 GB | 4.87s |
> | HPB | 5,431 G | 6.8 GB | 0.93s |
>
>
> *Inference - Forward Only:*
> | Model | FLOPs | VRAM | Wall Clock |
> | :--- | :--- | :--- | :--- |
> | Mamba | 1,452 G | 0.53 GB| 0.13s |
> | Hyena | 1,391 G | 1.1 GB | 0.26s |
> | Standard Transformer | 125,392 G | 28.7 GB | 19.30s |
> | Transformer + FlashAttn | 125,281 G | 2.4 GB  |1.30s |
> | HPB | 1,316 G | 1.2 GB | 0.29s |

---

### Official Review · Reviewer_R4oi · 2025-10-30

**Soundness:** 2
**Presentation:** 1
**Contribution:** 1
**Rating:** 2
**Confidence:** 5

**Summary:**

This paper proposes the Holistic Protean Block (HPB) for DNA sequence modeling, consisting of three components: (1) Locus Plasticity Layer (LPL) using dynamic multi-scale convolutions for local features, (2) Domain Plasticity Layer (DPL) combining global convolutions and wavelet transforms for time-frequency modeling, and (3) Saliency Plasticity Layer (SPL) for channel and position feature weighting. The authors claim HPB achieves "multi-level plasticity" to capture diverse DNA patterns including local motifs, long-range dependencies, and periodic signals. Evaluated on genomic benchmarks, the HPB-based model (HPB-DNA) reports state-of-the-art performance with quasi-linear complexity O(L log L).

**Strengths:**

1. Modeling DNA from a frequency perspective is novel and interesting, as DNA sequences indeed exhibit periodic patterns that current methods overlook. However, this novel point is obscured by excessive packaging of incremental architectural designs, diminishing the paper's clarity and impact.

2. The experimental evaluation is comprehensive, covering diverse benchmarks including regulatory annotation, histone markers, and variant effect prediction.

3. The ablation studies systematically validate the contribution of each component.

**Weaknesses:**

1. Most architectural components are incremental improvements: multi-scale convolutions, gated convolutions, and dual-axis feature gating are well-established techniques. While incremental engineering is acceptable, the authors unnecessarily complicate these straightforward designs with unclear terminology, making the paper difficult to read and understand.

2. The paper emphasizes "long-range DNA sequence modeling" yet pre-trains on only 1K bp sequences (Appendix A.2), far shorter than Enformer (196K bp). This length cannot capture enhancer-promoter interactions mentioned in the motivation nor demonstrate the claimed dynamic modeling capabilities. Moreover, when both pre-training and evaluation use short sequences (<= 1K bp), the O(L log L) complexity advantage over linear methods becomes negligible. This undermines the "long-range" claim and makes comparisons with methods pre-trained on longer sequences potentially unfair.

3. The abstract and introduction lack clarity from both biological and technical perspectives, obscuring rather than clarifying the contributions. I strongly recommend the authors use simpler, community-standard terminology rather than fancy-sounding phrases that obscure meaning. If the authors truly want to use these terms, they should provide clear explanations in context; otherwise, it will be very confusing for readers.  For examples:
   - Lines 14-15: "single/multi-domain" operations are undefined, what is "domain" here?
   - Line 17: "plasticity" is non-standard in both AI and genomics literature
   - Line 18: "Locus Plasticity Layer" uses "locus" (a genomics term for gene position) when "position" suffices
   - Line 85: "construction coefficients" is simply gating/weighting as in MoE but needlessly renamed
   - Line 86: "meta features" lacks clear definition
   - "Holistic Protean Block" uses uncommon terminology to convey only a general description of the model design

  4. The related work omits discussion of wavelet transforms and frequency-domain methods, despite frequency modeling being the paper's main novelty. Since most genomics researchers are unfamiliar with signal processing techniques, comprehensive background is essential.

5. Line 82 claims the HPB is a "scalable building block," but the largest model tested is under 2M parameters. Given the architectural complexity (three parallel modules with wavelets, FFT, and dynamic convolutions), it is unclear whether this design can scale up like foundation models in NLP or other genomics models.

6. It appears that different pre-trained models were used for different downstream tasks, which seems to contradict the concept of a foundation model. Additionally, the hyperparameters across downstream tasks vary drastically, raising concerns that the downstream performance may be attributed to hyperparameter tuning rather than fundamental improvements.

**Questions:**

1. Line 51 claims Transformers "lack strong inductive biases for local patterns" citing Enformer, but this requires stronger evidence or clarification of what Enformer actually states.

2. Have the authors considered applying this architecture to sequence-to-function modeling tasks like Enformer (predicting gene expression from long genomic sequences)? The motivation emphasizes long-range dependencies and frequency-sensitive patterns, which are most relevant in such tasks requiring long-context predictions. The current evaluation on short-sequence benchmarks cannot adequately demonstrate whether the proposed complexity genuinely improves biological modeling or merely provides marginal gains on oversimplified tasks.

---

> ### Author Response · Authors · 2025-11-21
> **Part 1 of the responses**
>
> ***We sincerely appreciate your valuable review comments and the effort you have put into them. We have addressed each of the questions you raised in detail. If anything is unclear, please inform us, and we will respond promptly.***
> >W1. Contribution and motivation of each component.
>
> ***Response:*** First, we would like to clarify that our core modules differ fundamentally from the off-the-shelf techniques you mentioned:
> *LPL:* The primary contribution of the LPL module is achieving **token-level** flexibility and adaptation, with a focus on generating context-aware convolutional patterns specific to each token. As described in the initial submission (Lines 210-211), this is realized by producing unique routing weights for each token, which are then used to combine the outputs of the convolutional kernel bases. This mechanism enables the effective extraction of variable local information in DNA sequences. Crucially, it fundamentally differs from standard multi-scale convolutions, which lack token-level dynamic adaptation. We have provided additional experimental proof demonstrating that LPL is more effective than other local motif extractors (please see A.11 of the revised paper).
> *DPL:* DPL implements adaptive feature extraction and fusion across **both time and frequency domains**. As detailed in Lines 76-80 of the initial submission, this design targets frequency-sensitive genomic signals like nucleosome periodicity. This is fundamentally different from the gated convolutions you referenced, which operate **solely in the time domain**. The benefits of fusing multi-domain features are evidenced by the frequency analysis in Figure 5(A) and the strong performance of DPL in Section A.12.
> Finally, we emphasize that **the core novelty of our work is the synergistic integration of LPL, DPL, and SPL into a unified "multi-level plasticity" framework**. This design addresses the specific complexity of DNA, which consists of local motifs, long-range dependencies, and periodic patterns (as stated in Lines 11-12 of the initial submission). Since no single existing architecture can capture all these features simultaneously, our holistic approach provides a necessary solution.
>
> >W2. Experiments on long-range DNA sequences.
>
> ***Response:*** We would like to clarify that Appendix A.2 of our initial submission **does not state** that our model was pre-trained exclusively on 1K-length sequences. On the contrary, our initial submission already included multiple long-sequence experiments as detailed in Section 4.2. These specifically include the Chromatin Profile Prediction and Variant Effect Prediction (VEP) tasks.
> For the VEP task, our model was trained and evaluated on sequences of **131k bps (as described in Line 393 and Appendix A.8 of the initial submission). This included a direct, head-to-head comparison against Enformer (Figure 3).** As shown in the results, our model (at 7.5M parameters) consistently outperforms Enformer (at 252M parameters) across various settings, despite being over 30x smaller.
> Furthermore, we have added experiments on the BEND [1] task, which involves input sequences up to 14,000 bps. The results, presented below, demonstrate that our model holds a clear and significant advantage (+15%) over other state-of-the-art methods on this new long-range sequence benchmark as well (please see A.15 for more details).
> Finally, we reiterate that our pre-training configurations across all benchmarks strictly adhered to prior works (e.g., Hyena, Caduceus, and ConvNova). Specifically, models were pre-trained on 1K sequence lengths for the Genomic Benchmark and Nucleotide Transformer Benchmark, while significantly longer sequence lengths (e.g., >100K) were utilized for long-range benchmarks such as Variant Effect Prediction (please see A.3, A.4 and A.8 of the initial submission). This rigorous approach **guarantees the fairness and validity of all comparative results**.
>
> *Performance comparison on Gene Finding benchmark (BEND)*
> | | NT-H (500M) | DNABERT-2 (117M) | GENA-LM (336M) | HyenaDNA (6.5M) |ConvNova (7.4M) |HPB-DNA (6.6M) |
> | :--- | :--- | :--- | :--- | :--- |:--- |:--- |
> | MCC | 0.41 | 0.43 | 0.52 | 0.35 | 0.55 | **0.68** |
>
> *references*
> [1] BEND: Benchmarking DNA language models on biologically meaningful tasks, ICLR 2024.

---

> ### Author Response · Authors · 2025-11-21
> **Part 2 of the responses**
>
> >W3. Clarification of Terminology in the Abstract and Introduction.
>
> ***Response:*** We would like to clarify that our initial submission already presented a multi-faceted argument covering both biological and theoretical motivations in the Abstract and Introduction of the initial submission. Key instances include the following:
> **Biologically**, we defined the problem scope in Lines 11-12 (initial submission) by highlighting the complex interplay of motifs and periodic patterns in DNA. We further biologically justified the motivation of our frequency-domain approach in Lines 77-78 (initial submission) by citing specific examples of vital periodic signals, such as *nucleosome positioning and codon periodicity*, which are often overlooked by current architectures.
> **Technically**, we critiqued existing methods in Lines 13-15 (initial submission), noting that their static or time-domain-only operations limit the flexibility required for DNA modeling. We then outlined our technical solution in Lines 107-110 (initial submission), explaining how the LPL, DPL, and SPL modules are specifically designed to address these limitations through token-specific extraction, time-frequency fusion, and dynamic saliency scoring.
> Regarding the questions you raised about our terminology choices, we provide detailed explanations and adjustments for each term below:
> 1. *"Domain":* We clarified in the initial submission (Lines 20-21) that "domain" refers to the "time" and "frequency" domains. To prevent any ambiguity, we have revised the term "single-domain" to "time-domain-only" in the Abstract (Line 14).
> 2. *"Plasticity":* This is a standard term in machine learning, often used in contrast to "stability" to describe a model's adaptability [1][2]. In our paper, we use "plasticity" to precisely describe our model's capacity for dynamic, input-specific adaptation and its ability to flexibly adjust its feature extraction process to diverse and variable local patterns.
> 3. *"Locus" (for LPL)*: "Locus" is indeed a common genomics term. We chose it because it carries a dual meaning: it implies a "positional" or "token-level" operation, while simultaneously evoking the module's function—to flexibly identify locally significant biological regions in a DNA sequence (which corresponds to the genomic meaning of "locus"). Instead, using "positional" would risk confusion with the "positional embedding layer" used in Transformers, which is not our intended meaning.
> 4. *"Construction Coefficients":* The LPL module constructs a specialized feature for each token by combining a set of basic (meta) features with token-specific custom parameters (routing weights). This mechanism is analogous to constructing a high-dimensional vector from a basis, which justified the initial term "construction coefficients". To improve readability for a wider audience, we have simplified this term to "weights" in the revised paper.
> 5. *"Meta Features":* The prefix "meta" is a standard term in machine learning (e.g., "meta-learning") used to denote something "foundational" or "fundamental." As defined in Lines 208-209 of the initial submission, "meta features" are the foundational outputs of the kernel bases branch (...applies each kernel k_i to E to produce a group of meta features). To make this explicit, we have added a clarifying sentence in the revised paper (Lines 223-224): "These are basic features used for token-level feature construction."
> 6. *"Holistic Protean Block":* We chose this name to be precise. "Holistic" is a common term in machine learning [3][4] signifying a "complete" or "all-encompassing" approach. "Protean," while less common, precisely describes **both flexibility and adaptability**—the ability to change form. Our model's core design philosophy is to provide flexibility and adaptation at multiple levels and from different perspectives (e.g., local, global, time&frequency domains) to effectively capture complex patterns of DNA. Therefore, the name "Holistic Protean Block" accurately encapsulates this central architectural characteristic.
>
> *references*
> [1] Understanding Plasticity in Neural Networks, ICML 2023.
> [2] Loss of plasticity in deep continual learning, Nature 2024.
> [3] Holistic deep learning, Machine Learning, 2024.
> [4] Holistic prediction of the pKa in diverse solvents based on a machine-learning approach, Angewandte Chemie, 2020.

---

> ### Author Response · Authors · 2025-11-21
> **Part 3 of the responses**
>
> >W4. Related works about frequency modeling.
>
> ***Response:*** To enhance accessibility for a broader audience, we have added a related work summary on time-domain and frequency-domain signal processing in the revised paper (Lines 128-138). The content is as follows:
> In the field of signal processing, the analysis of one-dimensional data has historically been divided into two core perspectives: time-domain and frequency-domain. Time-domain analysis operates directly on the sequence, using methods like digital filters (e.g., Savitzky-Golay [1]) and autoregressive models to capture local behavior. The central idea of these methods is to understand the signal's dynamics by analyzing its temporal dependencies. Frequency-domain analysis focuses on the signal's global and periodic structures; the FFT [2] identifies spectral components but loses temporal information. The STFT offers a time-frequency compromise via spectrograms, while the Wavelet Transform [3] provides a flexible multi-resolution analysis ideal for non-stationary signals. These classic paradigms inspire modern deep learning: CNNs act as learnable time-domain filters, while model like Hyena [4] leverage FFT for accelerating long convolutions.
>
> *references*
> [1] What is a Savitzky-Golay filter? IEEE Signal Processing Magazine, 2011.
> [2] In Fast Fourier Transform and Convolution Algorithms. Springer, 1981.
> [3] A theory for multiresolution signal decomposition: the wavelet representation. IEEE transactions on pattern analysis and machine intelligence, 2002.
> [4] HyenaDNA: Long-Range Genomic Sequence Modeling at Single Nucleotide Resolution, NeurIPS 2023.
>
> >W5. Scalibility of HPB.
>
> ***Response:*** To ensure a fair comparison across different tasks, we initially adopted model sizes ranging from ~500K to ~7M parameters. This aligned with the parameter budgets established by prior works (e.g., Hyena, Caduceus, and ConvNova), adhering to standard conventions in the field.
> To further demonstrate the scalability of our architecture, we scaled our model to 100M and 500M (0.5B) parameters and evaluated them on the Nucleotide Transformer Benchmark. As shown in the table below, the results reveal a clear scaling law: the 100M model achieves substantial performance gains across all tasks compared to the 1.9M variant, and the 500M model yields further improvements over the 100M model.
> Notably, our 500M model outperforms NT-v2 at a comparable parameter scale. This finding strongly validates that our proposed architecture is not only effective but also scales more favorably than alternative genomic foundation models.
>
> *Performance of HPB-DNA with different scales:*
> | | NT-V2 (500M) | HPB-DNA (1.9M)| HPB-DNA (100M) | HPB-DNA (500M) |
> | :--- | :---: | :---: | :---: |:---: |
> | Promoter all | 97.61 | 97.01 | 97.85| **98.23**|
> | Non-TATA | 97.53 | 96.88 | 97.98 | **98.56**|
> | TATA | 96.35| 96.59 | 98.02| **99.15**|
> | Enhancer | 56.62 | 58.69 | 59.96 | **61.29**|
> | Enhancer types | 45.95 | 50.87 | 52.88| **62.97**|
> | Splice Acceptor | 98.64 | 97.16 | 98.18| **99.06**|
> | Splice Donor |  98.86 | 97.79 | 98.76| **99.28**|
> | Splice All | 98.37 | 97.12 | 98.57| **99.61**|
> | H3 | 79.14 | 82.03 | 84.01| **85.22**|
> | H3K4me1 | 54.21 | 55.39 | 57.09| **59.21**|
> | H3K4me2 | 32.12 | 53.89 | 57.38| **60.06**|
> | H3K4me3 | 40.25 | 54.86 | 60.88| **66.92**|
> | H3K9ac |  56.62 | 64.50 | 66.29| **68.13**|
> | H3K14ac | 54.07 | 60.55 | 65.38| **71.29**|
> | H3K36me3 |  62.74  | 63.92 | 66.61| **69.32**|
> | H3K79me3 | 63.71 | 67.88 | 69.55| **72.98**|
> | H4 | 79.66  | 83.12 | 84.91| **85.85**|
> | H4ac | 49.52 | 56.91 | 61.29| **66.69**|

---

> ### Author Response · Authors · 2025-11-21
> **Part 4 of the responses**
>
> >W6. Why use different pre-trained models and hyperparameters on different tasks?
>
> ***Response:*** **About different pre-trained models:** We adopted different pre-trained models across different tasks to align with standard configurations in prior work and ensure a fair comparison. For instance, prior works like Hyena, Caduceus, and ConvNova typically utilize models with ~500K parameters for the Genomic Benchmark, ~2M for the Nucleotide Transformer Benchmark, and larger architectures (>6M) for the Chromatin Profile and Variant Effect Prediction tasks. Indeed, employing models of varying sizes for different scenarios is a **common strategy in the development of foundation models**. For instance, the GPT series includes distinct architectures like GPT-5.1-Instant and GPT-5.1-Thinking to optimize performance for specific use cases.
> **About different hyperparameters:** On the Genomic Benchmark, we employed a single, uniform set of hyperparameters for all constituent datasets, as detailed in Appendix A.3 of the initial submission. Our method's substantial advantages on this benchmark were achieved under this consistent setting.
> On the Nucleotide Transformer (NT) benchmark, we adhered to the established protocol used in prior works (e.g., Hyena, ConvNova, and CADUCEUS) by adjusting common hyperparameters, such as batch size and learning rate, for each task. Indeed, task-specific hyperparameter adjustment during the fine-tuning of foundation models is a **widely recognized standard practice**. For example, Section A.3 of the BERT paper [1] explicitly notes that "The optimal hyperparameter values are task-specific... For each task, we selected the best fine-tuning hyperparameters on the Dev set."
>
> *references*
> [1] BERT: Pre-training of Deep Bidirectional Transformers for Language Understanding, NAACL 2019.
>
> >Q1. Claims about Transformers.
>
> ***Response:*** We would like to point to the design rationale of the Enformer [1] model itself. The Enformer paper states (Methods, paragraph 4): "The convolution tower preceding MHA in the Enformer model serves to perform an analogous operation of embedding nucleotide segments and **contributes a compelling inductive bias** for adjacent nucleotides to function together in motifs." This description confirms that a key role of the convolution tower is to provide an inductive bias for local feature extraction, a capability where the core Transformer architecture is known to be less effective. In fact, this is a well-established consensus in the machine learning community. For instance, the Vision Transformer (ViT) paper [2] makes a similar observation (Introduction, paragraph 4) when comparing Transformers to CNNs: "This seemingly discouraging outcome may be expected: Transformers lack some of the inductive biases inherent to CNNs...".
>
> *references*
> [1] Effective gene expression prediction from sequence by integrating long-range interactions, Nature methods 2021.
> [2] An Image is Worth 16x16 Words: Transformers for Image Recognition at Scale, ICLR 2021.
>
> >Q2. Applying the model to gene expression prediction.
>
> ***Response:*** First, we would like to emphasize that our initial submission already provided substantial evidence from tasks involving both short- and long-range DNA sequences, achieving leading performance in both short- and long-range dependency modeling (see Section 4.1 and 4.2 of the initial submission). Specifically, our Variant Effect Prediction experiment (Figure 3 of the initial submission) was conducted on sequences up to 131k bps and was explicitly evaluated at different distances from the TSS. Notably, our model (at 7.5M parameters) consistently outperforms Enformer (at 252M parameters) across all settings, despite being over 30x smaller. To further validate this, we have also added new results on the BEND [1] task (up to 14,000 bps in length). The results, presented below, show that our method again holds a clear and significant advantage (please see A.15 for more details).
> While the primary focus of this paper is on foundational DNA sequence modeling, a choice made partly to ensure fair comparison with existing models that were evaluated solely on DNA sequence tasks, we agree that predicting gene expression from long genomic sequences is an important and interesting application. We greatly appreciate this suggestion and intend to explore it in future work.
>
> *Performance comparison on Gene Finding benchmark (BEND):*
> | | NT-H (500M) | DNABERT-2 (117M) | GENA-LM (336M) | HyenaDNA (6.5M) |ConvNova (7.4M) |HPB-DNA (6.6M) |
> | :--- | :--- | :--- | :--- | :--- |:--- |:--- |
> | MCC | 0.41 | 0.43 | 0.52 | 0.35 | 0.55 | **0.68** |
>
> *references*
> [1] BEND: Benchmarking DNA language models on biologically meaningful tasks, ICLR 2024.

---

> > ### Comment · Reviewer_R4oi · 2025-11-21
> >
> > Thank you for the effort, which addressed many of my concerns, especially regarding some terminology that I initially found confusing. The results after scaling up the model also look good. I have updated my score.
> >
> > However, I still have some concerns.
> >
> > **1. Short sequence pre-training**
> >
> > First, the authors have a misunderstanding of my concern.
> >
> > - I did not claim that the authors pre-trained **exclusively** on 1K sequences. The word “only” in my review “pre-trains on only 1K bp sequences” was to emphasize the sequence length is short under 'long-range' claims. It's a misunderstanding. In W6, I actually asked about why the authors used different pre-trained models.
> > - I apologize for the incorrect reference to A.2. The **1k pre-training length information** is actually in A.3 and A.4, not A.2. I confused the section numbers since A.2's title is **"Details of Pre-training"** and it's on the same page as A.3. I have corrected this in my review.
> >
> > My concern is not about whether the authors included long-sequence tasks with different pre-trained models, but about **the inconsistency between 1k pre-training and the paper's main motivation**. The motivation is reflected in such as the "long-range" in the paper title and the enhancer-promoter interaction in Figure 1. However, most of the experiments are based on the 1k length models. The authors can refer to my concern in Point 3 to address it.
> >
> > **2. Clarification of terminology**
> >
> > Thank you for further clarification. These explanations help me better understand your terminology.
> >
> > However, these terms are uncommon not only in genomic deep learning, but also in current mainstream AI areas such as LLMs. Although the authors listed some references in the response, I found they were not cited in the manuscript. I believe these terms would be very confusing for most readers of this paper.
> >
> > I still suggest that the authors either use more common terms, or explain their meaning when they first appear in the manuscript. I appreciate the authors have already modified the usage of "domain" in the abstract. But other problems remain. For example:
> > -  Your response mentioned that you critiqued "time-domain-only" in lines 13-15 of the initial submission. In fact, the initial version only had "single-domain", which is not defined.
> > - "Meta feature" appears on line 87 in the revised manuscript, but you said it's defined on line 208 in your response.
> > - I expect "Protean" means one model should work well on different sequence lengths. See my Point 3 for this concern.
> >
> > **3. Different pre-trained models and downstream hyperparameters**
> >
> > Thank you for clarifying that you followed previous work.
> >
> > But I still think this different pre-trained model practice is not good. I am concerned about whether a long-sequence pre-trained model can generalize well on short sequence downstream tasks. NT was pre-trained at 12k length only. This concern is consistent with my Point 1 in this response. **Since your model name contains "Protean", it should have the ability to model dynamically.** I expect long-sequence pre-training should also work well on short sequence downstream tasks.
> >
> > Besides, I am curious why genomic benchmark uses a fixed strategy, but NT benchmark uses different hyperparameters? NT's paper used only one set of hyperparameters.
> >
> >
> > **4. Minor: Citation errors in the authors' response**
> >
> > I would like to clarify that:
> > - BEND **was not published** in ICLR 2023.
> > - Enformer **was not published** in Nature.

---

> ### Author Response · Authors · 2025-11-24
> **First part of the further responses**
>
> ***We thank you for your follow-up comments and are glad to have the opportunity for further discussion. To address your questions, we have provided additional detailed explanations below, and we sincerely hope this fully clarifies your remaining concerns. We eagerly await your feedback.***
> *(Tables are on the second part of the further responses)*.
> > Points 1&3: About pre-training and hyperparameters.
>
> ***Response:***
> **About the pre-training settings:**
> We first apologize for the misunderstanding regarding your concern about pre-training setup. As stated in our previous response, our original submission included pre-training tasks using sequence lengths ranging from 1K up to 131K (Variant Effect Prediction) to maintain a fair comparison with prior works. However, we agree with your suggestion: pre-training on long sequences and then generalizing to shorter sequence tasks is a highly interesting and practically relevant setting.
> To explore this direction, we conducted additional experiments. We pre-trained our model on the hg38 DNA sequence using two different sequence lengths: 1K and 12K (while maintaining the same model size). We then evaluated both pre-trained models on the Genomic Benchmark and the Nucleotide Transformer Benchmark. The results are summarized below (T1 and T2).
> The results clearly indicate that our model pre-trained on 12K sequences generally outperforms the model pre-trained on 1K sequences, either exceeding or matching the performance across multiple datasets.
> We hypothesize that this improvement stems from the fact that pre-training on shorter sequences necessarily truncates long-range dependency information, which prevents the model from learning complete contextual associations. Utilizing a longer sequence length alleviates this issue, allowing the model to learn more accurate and holistic context-aware information. This outcome further validates that the **"protean" nature** of our architecture extends beyond mere feature pattern variability to encompass robustness and generalization across sequence lengths. These results have been incorporated into the revised paper (Appendix A.17).
> **About the hyperparameters:**
> We first address this point by considering the distinct characteristics of the Genomic Benchmark (GB) and the Nucleotide Transformer Benchmark (NT). The NT benchmark is inherently more challenging due to its higher task granularity (18 downstream tasks compared to 8 in GB), requiring models to focus on varied features. This disparity explains why task-specific hyperparameter tuning was necessary on NT to achieve maximal performance (a common practice followed by prior works, such as Hyena and Caduceus), whereas uniform hyperparameters proved sufficient for achieving state-of-the-art (SOTA) results on GB.
> However, we observed that this hyperparameter sensitivity diminishes as the model scale increases. Specifically, we conducted an experiment on our 500M model (comparable in size to NT-v2) using both differentiated (task-specific) and unified (learning rate=1e-4, batch size=256) hyperparameter settings on the NT benchmark. As shown in the results below (T3), the performance difference between the two configurations is negligible at the 500M scale. We hypothesize this is because larger models possess greater representational capacity and robustness, which significantly reduces their sensitivity to hyperparameters compared to smaller variants.
>
> > Point 2: Clarification of terminology.
>
> ***Response:*** Thank you for the suggestions. To enhance readability for the readers, we have incorporated the relevant literature cited in our responses into the revised paper (Lines 83-84 and 95-96). Furthermore, regarding the specific examples you raised, we provide the following clarifications and adjustments:
> *"Domain":* We clarified in our previous response that the term 'domain' was further discussed upon in Lines 20-21 of the initial submission. To prevent any potential ambiguity, we have now implemented a more precise definition at its first mention in the Abstract (Lines 14-15 of the revised paper) by clarifying the context as "time-domain-only signal processing operations".
> *"Meta features":* We have provided a more detailed explanation of this concept in the revised paper (Lines 86-87).
> *"Protean":* Please refer to our detailed rationale for this term, which is provided in the response of "Points 1&3".
>
> > Point 4: Minor citation errors in the responses.
>
> ***Response:*** We have revised the citations of "BEND" (ICLR 2023 -> ICLR 2024) and "Enformer" (Nature -> Nature methods) in our responses, thanks for the reminder.

---

> ### Author Response · Authors · 2025-11-24
> **Second part of the further responses**
>
> *T1: Ablation of pre-training length on Genomic Benchmark:*
> | Model| HPB-DNA (1K length)|HPB-DNA (12K length)|
> | :---:| :---: | :---: |
> | Mouse Enhancers | 82.38 | 84.15|
> | Coding vs. Intergenomic | 94.46 |  94.49|
> | Human vs. Worm | 97.19 |  97.16|
> | Human Enhancers Cohn | 74.96 |  75.23|
> | Human Enhancers Ensembl | 91.08 |  91.68|
> | Human Regulatory | 94.20 |  94.26|
> | Human Nontata Promoters | 94.78 | 94.98 |
> | Human OCR Ensembl | 81.95 |  83.01|
>
>
> *T2: Ablation of pre-training length on Nucleotide Transformer Benchmark:*
> | Model | HPB-DNA (1K length) | HPB-DNA (12K length)  |
> | :--- | :---: | :---: |
> | Promoter all | 97.01 |    97.65 |
> | Non-TATA | 96.88 |    97.16|
> | TATA | 96.59 |   97.08 |
> | Enhancer | 58.69 |   59.31 |
> | Enhancer types | 50.87 |   52.47 |
> | Splice Acceptor | 97.16 |    97.52|
> | Splice Donor | 97.79 |   97.76 |
> | Splice All | 97.12 |   97.68 |
> | H3 | 82.03 |  83.11  |
> | H3K4me1 | 55.39 |   56.19 |
> | H3K4me2 | 53.89 |   53.95 |
> | H3K4me3 | 54.86 |   55.09 |
> | H3K9ac | 64.50 |   64.82 |
> | H3K14ac  | 60.55 |   61.33 |
> | H3K36me3 | 63.92 |   64.28 |
> | H3K79me3  | 67.88 |   68.19 |
> | H4 | 83.12 |   83.36 |
> | H4ac | 56.91 |  57.87  |
>
>
> *T3: Performance of HPB-DNA with different hyperparameter settings:*
> | Model | HPB-DNA (500M, different hyperparameters) | HPB-DNA (500M, unified hyperparameters) |
> | :--- | :---: |:---: |
> | Promoter all | 98.23|  98.31|
> | Non-TATA | 98.56|  98.54|
> | TATA | 99.15|  99.21|
> | Enhancer | 61.29|  61.39|
> | Enhancer types | 62.97|  62.93|
> | Splice Acceptor | 99.06|  99.08|
> | Splice Donor | 99.28|  99.30|
> | Splice All | 99.61|  99.59|
> | H3 | 85.22|  85.22|
> | H3K4me1 | 59.21|  59.46|
> | H3K4me2 | 60.06|  60.10|
> | H3K4me3 | 66.92|  66.90|
> | H3K9ac | 68.13|  68.15|
> | H3K14ac | 71.29|  71.38|
> | H3K36me3 | 69.32|  69.66|
> | H3K79me3 | 72.98|  73.01|
> | H4 | 85.85| 85.82 |
> | H4ac | 66.69|  66.73|

---

> ### Comment · Reviewer_R4oi · 2025-11-26
>
> Thank the authors for addressing my concerns about different models and different hyperparameters to some extent. I suggest the authors also use the same model and hyperparameters in the final version to make the results more convincing.
>
> I will keep my overall assessment, with a slight increase in the specific score. This is my final decision.
>
> 1. I am not satisfied with the presentation of this paper.
> - As mentioned in my initial review, considering the periodicity of DNA is somewhat interesting for me. If the authors go deeper into this aspect and observe that it brings stable improvements, I would be more satisfied. However, the authors stacked other relatively boring modules, which are essentially engineering tricks that cannot bring me new insights. Reviewer hALi shares the same view on this point.
> - I don't like that the authors use so many terminologies rarely used in the literature without explanation, and most of them only convey a very general meaning, which means removing these terminologies would not hurt the understanding of the specific implementation at all.
>
> 2. The experimental results are not convincing
> - I personally do not care about benchmarks like Genomic Benchmark where naive CNN baselines can already perform well. The improvements within the margin of error on the NT benchmark are also not convincing. Most results in Table 2 have improvements less than one standard deviation, but the authors can still mark the highest mean in most entries, which raises concerns about cherry-picking results. Not annotating the specific versions of models in the initial version can easily mislead readers. The authors reports results without standard deviations during rebuttal, which may be improvements within the margin of error caused by cherry-picking. Some baseline results are reported much lower for unknown reasons. For example, the NT-V2 results on H3K4me2 reported by the authors in Table 2 are far lower than the results reported in the NT paper.
> - eQTL variant effect prediction is a relatively meaningful task in my opinion, but the baselines for this task are too weak. Please check the results in the original papers of Enformer, Borzoi, and AlphaGenome.

---

> ### Author Response · Authors · 2025-11-26
>
> Thanks for your feedback. Regarding your points, we wish to emphasize the following key aspects of our work:
> ***Presentation and Clarity***
> * As detailed in our previous response to your "W1" concern, the core modules of our approach (specifically the LPL and the DPL) **differ fundamentally** in mechanism from existing models. Furthermore, they demonstrate **notable performance advantages**. We dedicated Section 4.3 of our paper to a multi-faceted comparison of LPL, DPL, and SPL against other state-of-the-art operators, clearly showcasing the superiority of our modules. Crucially, **the novelty of our work lies in the synergistic integration of LPL, DPL, and SPL into a unified "multi-level plasticity" framework.** This holistic approach is necessary to address the specific complexities of DNA sequences, as no single existing operator can simultaneously capture all required features.
> * As we explained and substantiated in our previous response to "W3", the majority of the terminology we use is standard within the fields of machine learning or bioinformatics. Our description of the proposed method is both accurate and necessary. We have, in addition, provided specific explanations and simplifications for certain terms, and all these clarifications will be incorporated into the final version of the manuscript.
>
> ***Experimental Rigor and Results***
> We must first stress that Section 4.1 of our paper explicitly states that all experimental results are the average and standard deviation derived from **multiple runs using different random seeds**. The results presented in this rebuttal maintain the same statistical rigor. Therefore, our findings are **statistically accurate and stable**, completely **ruling out the possibility of "cherry-picking."**
> Regarding the performance of NT-V2, as explicitly stated in the caption of our Table 2, we utilized the official pre-trained model and recommended parameters provided by the NT-V2 authors for our fine-tuning process. We ran the fine-tuning for ten iterations and reported the average performance. The discrepancy between the reproduced performance and the results presented in the original NT-V2 paper is a universally observed phenomenon in prior works (e.g., Hyena, Caduceus, and ConvNova). Furthermore, our reproduced NT-V2 results are highly consistent with those reported in these previous studies. E.g., for the H3K4me2 you mentioned, we reproduced the NT-V2 result at 32.12, which is very close to the 31.9 reproduced by Caduceus. Therefore, our reported NT-V2 results are reliable and establish a fair baseline for comparison with our proposed method. Besides, as we stated in the Abstract, we'll release the code for reproduction.
> About the tasks you mentioned:
> *Nucleotide Transformer (NT) Benchmark:* Statistically, the mean accuracy of our method surpasses the current SOTA, and it exhibits a relatively smaller overall standard deviation. This **statistical evidence** sufficiently demonstrates our advantage in both accuracy and stability on this complex task.
> *Genomic Benchmark (GB):* While the complexity of GB is lower than NT, it remains a fundamental and widely used benchmark. Our method achieves an improvement of up to 5.75% over the SOTA on this benchmark. Achieving such a high gain on GB is non-trivial (newer methods like Caduceus, ConvNova, and xLSTM typically show performance margins within 2% on this benchmark), which strongly validates the efficacy of our approach.
> *Variant Effect Prediction (VEP):* For this task, we adopted the latest experimental settings established in the DNA foundational model community (see Caduceus). We then performed fair, same-setting comparisons against both advanced DNA foundational models and classic bioinformatics models. Our model surpassed the performance of all competing methods across different settings, providing sufficient evidence for the effectiveness of our architecture in ultra-long sequence tasks.
> Finally, we emphasize that, as stated in our initial submission, all our experimental settings strictly adhere to prior works to ensure fair and rigorous comparison. Furthermore, the specific model size was clearly annotated in the table for each task, and detailed descriptions of the parameters and training configurations for all models were provided in the Appendix. In conclusion, we reiterate the notable advantages our model has already demonstrated:
>   * Up to 5.75% improvement over the SOTA on the Genomic Benchmark.
>   * Up to 1.81% gain over the SOTA on the challenging Nucleotide Transformer Benchmark.
>   * Achieving a lead under all settings of ultra-long sequence tasks like Variant Effect Prediction.
>   * A 13% performance gain over the SOTA on BEND.
>
> These tasks collectively span both short and ultra-long sequences, definitively proving our model's generalization capability across **various sequence lengths** and its superior accuracy compared to alternative methods.

---

### Official Review · Reviewer_A7YH · 2025-11-01

**Soundness:** 3
**Presentation:** 2
**Contribution:** 2
**Rating:** 4
**Confidence:** 3

**Summary:**

This paper introduces a new architecture that integrates three different functional layers (LPL, DPL, SPL) that work together to learn a holistic representation of DNA sequences. The HPB-DNA model demonstrates strong empirical performance, setting new STOA results on genomic benchmarks with efficient, quasi-linear scaling.

**Strengths:**

- Formulating DNA analysis in terms of multi-level plasticity (locus, domain, saliency) is novel and seems like a good way to conceptualize genomic data.

- The authors evaluate across multiple genomic datasets and show consistent improvements under comparable model sizes.

- The quasi-linear complexity and small parameter count appear to be a promising way to address genomics tasks.

**Weaknesses:**

- The paper introduces several novel architectural operations, but there is not enough work done in the current ablation study isolating their individual contributions. It would strengthen the work to include experiments showing how the model performs when each new component is removed or replaced.
- The paper does not clearly articulate the theoretical or biological motivation behind combining a global convolution with a wavelet transform. The rationale for the wavelet path, in particular, should be clarified - what specific objective does it serve? Providing stronger domain-grounded justification would make the design choice more compelling.
- The paper asserts that HPB is scalable to foundation-model scale, but empirical evidence is limited to relatively small (< 5M parameter) models. The authors should demonstrate that HPB is actually scalable in a way that improves model performance.
- Table 4 presents performance on a limited subset of datasets, which may give the impression of selective reporting. Showing results across all genomic benchmarks would provide a more comprehensive and fair evaluation of model performance. Similarly, the same issue is present in Table 5.
- Given the model’s efficiency claims, the paper should include a quantitative comparison of computational cost across methods, covering memory usage, wall clock training and inference time, and FLOPs to provide a comprehensive assessment of efficiency.

**Questions:**

- How sensitive is the model to changes in the number of wavelet decomposition levels and to kernel-base selection in the LPL? In general, further reporting on how hyperparameters were derived would be helpful.

- Can the authors quantify how much each plasticity layer contributes to the overall improvement?

- Are there domain-related failure modes where multi-domain modeling introduces noise or instability (such as with sequences with no clear periodicity)?

- Could the authors provide additional analyses on whether the learned features are biologically interpretable?

---

> ### Author Response · Authors · 2025-11-21
> **Part 1 of the responses**
>
> ***We sincerely appreciate your valuable review comments and the effort you have put into them. We have addressed each of the questions you raised in detail. If anything is unclear, please inform us, and we will respond promptly.***
> >W1&Q2. Detailed ablation of each plasticity layer.
>
> ***Response:*** In our initial submission, we validated the individual efficacy of each component through comparative experiments against alternative methods (see Table 4 for LPL, Table 5 for DPL, and Figure 4 for SPL).
> To further clarify the specific contribution of each component, we have conducted additional ablation studies. As shown in the table below, removing any single module leads to a decline in overall performance. This confirms that the LPL, DPL, and SPL are structurally and functionally complementary; their synergistic integration is essential for achieving optimal results. These results have been incorporated into the updated Table 4 of the revised paper.
>
> *Ablation of each component of HPB:*
> || w/o LPL|w/o DPL | w/o SPL | HPB |
> | :---:| :---: | :---: | :---:| :---: |
> | Mouse Enhancers |79.39 | 79.21 | 81.02| **82.38** |
> | Coding vs. Intergenomic |91.06 | 89.98| 92.11 | **94.46** |
> | Human vs. Worm |94.67 | 90.01 | 95.22| **97.19** |
> | Human Enhancers Cohn |71.19 | 69.53| 72.01 | **74.96** |
> | Human Enhancers Ensembl | 88.66| 85.91| 90.29 | **91.08** |
> | Human Regulatory | 90.34| 89.19 | 91.36| **94.20** |
> | Human Nontata Promoters | 91.05| 88.35| 93.19 | **94.78** |
> | Human OCR Ensembl | 78.61| 76.20|79.38 | **81.95** |
>
> >W2. Motivation behind the Global Convolution and Wavelet Paths.
>
> ***Response:*** We elaborate on the motivation of combining global convolution and a wavelet transform, viewed separately from the biological and theoretical standpoints.
> **Biologically**, DNA sequences encompass not only linear, time-domain information (such as local motifs and long-range dependencies) but also crucial frequency-domain signals often associated with specific biological properties [1], most notably the ~10 bp periodicity of nucleosome positioning [2] and ~3 bp for protein coding [1].
> **Theoretically**, models operating solely in the time domain often lack the intrinsic mechanisms to capture these periodic patterns effectively. The specific objective of our wavelet path is to perform multi-resolution analysis, decomposing the sequences into distinct frequency sub-bands for feature extracting. This enables the model to isolate and explicitly process these biologically vital periodic features that might otherwise remain obscured [3].
> We initially discussed this rationale in the Introduction (Lines 76-80) and Related Works (Lines 158-161) of the initial submission. We have now added a supplementary theoretical explanation to the Related Works section (Lines 134-136) of the revised paper.
> Crucially, we emphasize that these two paths are designed to be complementary. The global convolution path efficiently captures global dependencies in the sequential (time) domain, while the wavelet path provides a spectral (multi-resolution frequency) perspective. This synergy results in a holistic representation, a conclusion that is strongly justified by our analysis in Section 4.4 (Fig. 5A), which demonstrates that our model (unlike its predecessors) successfully identifies the known biological periodicity in nucleosome-bound DNA.
>
> *references*
> [1] Comparative analysis of periodicity search methods in DNA sequences, Computational Biology and Chemistry, 2014.
> [2] On the Relationship between DNA Periodicity and Local Chromatin Structure, In Annual International Conference on Research in Computational Molecular Biology, 2009.
> [3] Wavelet analysis of DNA sequences. Phys. Rev. E, 1996.

---

> ### Author Response · Authors · 2025-11-21
> **Part 2 of the responses**
>
> >W3. Scalability of HPB.
>
> ***Response:*** To ensure a fair comparison across different tasks, we initially adopted model sizes ranging from ~500K to ~7M parameters. This aligned with the parameter budgets established by prior works (e.g., Hyena, Caduceus, and ConvNova), adhering to standard conventions in the field.
> To further demonstrate the scalability of our architecture, we scaled our model to 100M and 500M (0.5B) parameters and evaluated them on the Nucleotide Transformer Benchmark. As shown in the table below, the results reveal a clear scaling law: the 100M model achieves substantial performance gains across all tasks compared to the 1.9M variant, and the 500M model yields further notable improvements over the 100M model.
> Notably, our 500M model outperforms NT-v2 at a comparable parameter scale. This finding validates that our proposed architecture is not only effective but also scales more favorably than alternative genomic foundation models.
>
> *Performance of HPB-DNA with different scales:*
> | | NT-V2 (500M) | HPB-DNA (1.9M)| HPB-DNA (100M) | HPB-DNA (500M) |
> | :--- | :---: | :---: | :---: |:---: |
> | Promoter all | 97.61 | 97.01 | 97.85| **98.23**|
> | Non-TATA | 97.53 | 96.88 | 97.98 | **98.56**|
> | TATA | 96.35| 96.59 | 98.02| **99.15**|
> | Enhancer | 56.62 | 58.69 | 59.96 | **61.29**|
> | Enhancer types | 45.95 | 50.87 | 52.88| **62.97**|
> | Splice Acceptor | 98.64 | 97.16 | 98.18| **99.06**|
> | Splice Donor |  98.86 | 97.79 | 98.76| **99.28**|
> | Splice All | 98.37 | 97.12 | 98.57| **99.61**|
> | H3 | 79.14 | 82.03 | 84.01| **85.22**|
> | H3K4me1 | 54.21 | 55.39 | 57.09| **59.21**|
> | H3K4me2 | 32.12 | 53.89 | 57.38| **60.06**|
> | H3K4me3 | 40.25 | 54.86 | 60.88| **66.92**|
> | H3K9ac |  56.62 | 64.50 | 66.29| **68.13**|
> | H3K14ac | 54.07 | 60.55 | 65.38| **71.29**|
> | H3K36me3 |  62.74  | 63.92 | 66.61| **69.32**|
> | H3K79me3 | 63.71 | 67.88 | 69.55| **72.98**|
> | H4 | 79.66  | 83.12 | 84.91| **85.85**|
> | H4ac | 49.52 | 56.91 | 61.29| **66.69**|
>
> >W4. Some datasets are missing in Table 4 and 5.
>
> ***Response:*** Due to space constraints in the initial submission, we selected a representative subset of datasets to demonstrate our comparative analysis. Following your recommendation, we have expanded the ablation study to include all datasets within the Genomic Benchmark and Nucleotide Transformer Benchmark and have updated Table 4 and Table 5 accordingly (shown below).
> As evident in these updated results, the performance advantages of our LPL and DPL modules over competing architectures hold consistently across the full suite of datasets. This provides a more comprehensive validation of our method's superiority and demonstrates its strong generalization capabilities. We have incorporated these detailed results into Appendix A.11 and A.12 of the revised paper.
>
> *Ablation of local motif extractors on Genomic Benchmark:*
> || CondConv|ConvNeXt | MambaOut | GCB | LPL |
> | :---:| :---: | :---: | :---:| :---: | :---: |
> | Mouse Enhancers |79.89 | 80.22 | 78.06| 80.38 | **82.38** |
> | Coding vs. Intergenomic |92.11 | 92.38 | 93.19| 93.23 | **94.46** |
> | Human vs. Worm |95.38 | 96.12 | 95.87 |96.29 | **97.19** |
> | Human Enhancers Cohn |73.39 | 72.58 |73.01| 73.03 | **74.96** |
> | Human Enhancers Ensembl | 89.66| 87.85 | 88.03| 87.29 | **91.08** |
> | Human Regulatory | 92.37| 91.99 | 92.96| 91.20 | **94.20** |
> | Human Nontata Promoters | 94.12| 94.20 | 94.13| 94.32 | **94.78** |
> | Human OCR Ensembl | 79.95| 80.81 | 80.35 |80.11 | **81.95** |
>
> *Ablation of global dependency capturing on Nucleotide Transformer Benchmark:*
> | | long conv (Hyena) | self attention (Transformer) | DPL |
> | :--- | :---: | :---: | :---: |
> | Promoter all | 95.73 | 96.02 | **97.01** |
> | Non-TATA | 95.12 | 95.23 | **96.88** |
> | TATA | 96.03 | 96.28 | **96.59** |
> | Enhancer | 53.10 | 52.98 | **58.69** |
> | Enhancer types | 48.66 | 47.98 | **50.87** |
> | Splice Acceptor | 95.99 | 96.36 | **97.16** |
> | Splice Donor | 96.39 | 96.88 | **97.79** |
> | Splice All | 95.82 | 94.11 | **97.12** |
> | H3 | 79.86 | 78.55 | **82.03** |
> | H3K4me1 | 54.02 | 53.11 | **55.39** |
> | H3K4me2 | 51.93 | 53.76 | **53.89** |
> | H3K4me3 | 50.32 | 49.39 | **54.86** |
> | H3K9ac | 60.89 | 59.45 | **64.50** |
> | H3K14ac | 56.91 | 57.87 | **60.55** |
> | H3K36me3 | 62.08 | 62.66 | **63.92** |
> | H3K79me3 | 65.31 | 66.85 | **67.88** |
> | H4 | 78.27 | 79.33 | **83.12** |
> | H4ac | 56.56 | 55.99 | **56.91** |

---

> ### Author Response · Authors · 2025-11-21
> **Part 3 of the responses**
>
> >W5. Model efficiency of HPB.
>
> ***Response:*** While our initial submission provided a theoretical analysis of computational complexity (Section 3.2), we have now followed your recommendation to conduct a comprehensive empirical evaluation of FLOPs, memory footprint, and runtime.
> We performed a fair comparison against Mamba, Hyena, a standard Transformer, and a Transformer with Flash Attention. All models were configured with 12M parameters and evaluated on input sequences of 100k length. Training and inference tests for a single sample were conducted on the Nvidia A800 GPU.
> As shown in the results below, our model's FLOPs are **significantly lower** than the Transformer's and are even slightly lower than those of Hyena and Mamba. This empirically validates the quasi-linear computational complexity of our approach.
> Regarding memory usage and wall-clock time, our model is significantly more efficient than the Transformer, though currently slightly higher than Mamba and Hyena. It is crucial to note that architectures like Transformer (Flash Attention) benefit from highly mature, engineering-level accelerations. Our current implementation has not yet undergone such low-level optimization. Given our lower FLOPs, this indicates substantial potential for further acceleration, which we prioritize as future work.
> In summary, our model surpasses the Transformer in efficiency and remains competitive with Mamba and Hyena. Considering our model simultaneously achieves better accuracy on both short- and long-sequence tasks, we believe it represents a highly promising and efficient architectural alternative. These results have been included in the revised paper (please see A.14).
>
> *Training - Forward + Backward:*
> | Model | FLOPs | VRAM | Wall Clock |
> | :--- | :--- | :--- | :--- |
> | Mamba | 5,619 G | 5.8 GB | 0.46s |
> | Hyena | 5,573 G | 6.2 GB | 0.89s |
> | Standard Transformer| 517,328 G | OOM | N/A |
> | Transformer + FlashAttn | 516,931 G | 7.8 GB | 4.87s |
> | HPB | 5,431 G | 6.8 GB | 0.93s |
>
>
> *Inference - Forward Only:*
> | Model | FLOPs | VRAM | Wall Clock |
> | :--- | :--- | :--- | :--- |
> | Mamba | 1,452 G | 0.53 GB| 0.13s |
> | Hyena | 1,391 G | 1.1 GB | 0.26s |
> | Standard Transformer | 125,392 G | 28.7 GB | 19.30s |
> | Transformer + FlashAttn | 125,281 G | 2.4 GB  |1.30s |
> | HPB | 1,316 G | 1.2 GB | 0.29s |
>
> >Q1. Ablation of hyperparameters such as wavelet decomposition levels and kernel-bases.
>
> ***Response:*** We would like to clarify that ablation studies on both wavelet decomposition levels and kernel-bases selection were included in the appendix of our initial submission (please see Sections A.5 and A.6).
> In general, our findings show that while larger decomposition levels and a greater number of kernel bases can offer accuracy improvements, they also introduce more computational overhead. We therefore sought a balance between performance and efficiency. The selected values (a decomposition level of 3 and kernel bases of [1, 3, 5, 7]) represent this trade-off, providing a robust, general-purpose configuration.
>
> >Q3. Are there domain-related failure modes?
>
> ***Response:*** We respectfully submit that this is not the case. On the contrary, our model is capable of **effectively avoiding such domain-related failure modes**. The reasons are as follows:
> As illustrated in Figure 2, the DPL module processes the decomposed low-frequency and high-frequency components using both a global convolution and learnable scale coefficients, both of which are fully trainable. In cases where the input data or task is frequency-agnostic (i.e., frequency information is irrelevant or noisy), the model learns via gradient backpropagation to optimize these parameters to attenuate the contribution of the frequency-domain features. This mechanism effectively prevents unnecessary noise from impacting the final representation.
> This is precisely the key differentiator of our DPL module compared to traditional methods that operate inflexibly in either the time or frequency domain. Our model adaptively fuses time-frequency features, extracting the information most salient to the current task.

---

> ### Author Response · Authors · 2025-11-21
> **Part 4 of the responses**
>
> >Q4. Additional biologically interpretable analyses.
>
> ***Response:***  First, we would like to emphasize that Figures 5A and 5B of the initial submission already presented robust evidence of biological interpretability. Figure 5A demonstrated our model's ability to accurately capture the characteristic ～10bp periodic feature commonly observed in nucleosome sequences. Furthermore, Figure 5B showcased the model's success in learning high-response features with clear biological meaning (e.g., promoter regions) under an unsupervised training paradigm.
> Following your suggestion, we have now conducted additional case analysis experiments from both statistical and visualization perspectives to further support these findings. Specifically, we evaluated the predictive capability of individual sparse autoencoder (SAE) features. We utilized the activation of specific sparse features (i.e., non-zero values) to classify distinct gene regulatory elements (promoters, exons, CDS) and calculated their F1 scores. The results are summarized in the table below (where Transformer, Hyena, and HPB are trained using the same setting). **The leading performance of HPB provides compelling statistical evidence that our model systematically captures biologically meaningful patterns.**
> To rigorously validate these associations, we further performed a statistical comparison against chromosome-matched background controls. We found that feature activation intensity and frequency were significantly higher in target regions compared to controls ($p < 1e-300$). This robust statistical evidence confirms that our model extracts biological patterns, not random predictions.
> Finally, to complement this quantitative analysis, we have provided additional visualizations corresponding to CDS, shown in Figure 7 in the revised paper. The combination of quantitative metrics (F1 scores and statistical tests) and additional visualizations conclusively demonstrates the strong biological interpretability of our learned features. These new findings have been incorporated into the revised paper (please see A.16).
>
> *F1 score between features and genomic regions:*
> |  | Transformer (12.6M) | Hyena (12.8M) | HPB (12.3M) |
> | :--- | :--- | :--- | :--- |
> | promoter | 0.0601 | 0.0900 | **0.1320** |
> | exons | 0.0758 | 0.0830 | **0.0984** |
> | cds| 0.0341 | 0.0500 | **0.0719** |

---

> > ### Comment · Reviewer_A7YH · 2025-11-27
> >
> > Thank you to the authors for the extensive follow up experiments. Overall, many of my concerns were addressed. However, a few items remain:
> >
> > 1. Regarding the new Table 5 (performance comparison of local motif extractors), I’m curious why you substituted the convolution and dilated convolution models with CondConv and GCB, especially when it seems that conv and dilated conv models seem to be outperforming CondConv and GCB in some instances (Human vs. Worm). Also it seems like there is a discrepancy in reporting - Human vs. Worm LPL performance was originally 97.19 and Human Regulatory LPL 94.20 originally, but in the updated version it seems that the performances for LPL are now 94.46 and 97.19, respectively. Please clarify.
> >
> > 2. For Q3, could you provide an experiment (perhaps using synthetically generated sequences) to validate how the model effectively avoids domain-related failure modes?

---

> > > ### Author Response · Authors · 2025-11-27
> > > **Part 2/2 of the further responses**
> > >
> > > *Performance comparison on sequences with or without periodic signal interference (human_enhancers_cohn):*
> > > |  | Accuracy (%) | Feature Response Ratio |
> > > | :--- | :--- | :--- |
> > > | with periodic signal | 75.36 | 1218.03 |
> > > | without periodic signal | 75.29 | 1206.01 |
> > >
> > > *Performance comparison on sequences with or without periodic signal interference (human_nontata_promoters):*
> > > |  | Accuracy (%) | Feature Response Ratio |
> > > | :--- | :--- | :--- |
> > > | with periodic signal | 95.58 | 679.38 |
> > > | without periodic signal | 95.62 | 682.59 |

---

> > > > ### Comment · Reviewer_A7YH · 2025-11-27
> > > >
> > > > Thank you to the authors for the additional explanation and analysis. One final request (that other reviewers have also brought up). Could you please provide the standard deviations for all of your experimental results?

---

> > > > > ### Author Response · Authors · 2025-11-28
> > > > >
> > > > > We are glad to see that our additional explanation and analysis were well-received. Following your final suggestion, we have updated the revised paper to include standard deviations for all reported results, covering benchmarks, ablation studies, and extended experiments. Thank you for your valuable insights, which have significantly strengthened the quality of our work!

---

> ### Author Response · Authors · 2025-11-27
> **Part 1/2 of the further responses**
>
> ***We thank you for your follow-up response and are glad to have the opportunity for further discussion.***
> *(The additional experiment results are in the 'Part 2/2 of the further responses')*.
> **Response to Point 1:**
> * We updated the comparisons from "conv" and "dilated conv" to specific "CondConv" and "GCB" to ensure our evaluation included the most recent and relevant local motif extractors. As detailed in the table within our previous response and Table 10 of the revised paper, we have also expanded our evaluation to include models such as ConvNeXt and MambaOut. Crucially, we wish to clarify that the "dilated convolution" reported in our initial submission is precisely the GCB module from ConvNova. We respectfully refer you to our detailed response to Reviewer hALi (Q2) for specific clarification on this point.
> * Regarding your observation that Conv outperforms CondConv sometimes, our experiments confirm this phenomenon. We attribute this to the nature of CondConv's sample-level dynamics: it generates a single kernel for an entire sample, applying it uniformly across all positions. In DNA sequences, however, features vary drastically across different positions within the same sample (intra-sample heterogeneity). This creates a **"granularity mismatch"** where a single dynamic kernel cannot capture diverse local patterns. Conversely, a fixed convolution may perform better because applying a consistent kernel across samples can actually facilitate the learning of common motif patterns (inductive biases) shared among different samples, thereby enhancing performance to a certain extent. In contrast, our LPL module provides **token-level dynamics**, generating unique kernel weights for each specific position within a sample. This design effectively **resolves the granularity mismatch** inherent in CondConv. Furthermore, LPL allows the model to learn fine-grained motif inductive biases across different samples (via token-specific adaptation mechanisms), thereby achieving better performance compared to both standard Conv and CondConv. We have also included the full results of the 'conv' module in the revised paper (Table 10).
> * Regarding the discrepancy in results: We sincerely apologize for the alignment error in Table 5 of the revised paper. We confirm that the full ablation results provided in our previous response (which correspond to Table 10 in the appendix of the revised paper) are correct and consistent with the results in our initial submission. During the revision process, to conserve space in the main text, we included only a subset of the full LPL ablation results (specifically, the first three rows) in Table 5. However, we inadvertently retained the full list of dataset labels from the initial submission, leading to a misalignment between the data and the row headers. We have corrected these typographical errors and conducted a thorough proofreading of the entire manuscript. We are grateful for your attention to detail in pointing this out.
>
> **Response to Point 2:**
> 1. Following your suggestion, we conducted additional verification experiments to validate this behavior. We constructed contrastive groups (with vs. without periodic signals) derived from the human_enhancers_cohn and human_nontata_promoters datasets (Genomic Benchmark). In the groups containing periodic signals, we appended a 500-token random periodic signal (period = 5bp) to the end of each sequence sample. In the groups without periodic signals, we appended 500 non-periodic random bases. We then trained and evaluated the model on these datasets, where the objective remained enhancer or promoter prediction (independent of the appended signals).
> 2. We evaluated both classification accuracy and the feature response ratio (defined as the ratio of feature activation intensity in the target region relative to the padding region), reporting the mean results over 5 independent runs. As shown in the tables below, the test accuracies on both groups are highly comparable. Crucially, for the group with periodic signals, the feature response ratio is extremely high and aligns closely with that of the group without periodic signals. These results demonstrate that, for sequences either with or without periodic signals, our model effectively suppresses regions irrelevant to the prediction target rather than introducing multi-domain noise. This capability is largely attributed to the DPL module's mechanism for the dynamic scaling and fusion of time- and frequency-domain components. We have included these details in Appendix A.18 of the revised paper. Thank you for this insightful suggestion.
>
> *We hope that our further clarifications and supplementary experiments have adequately addressed your concerns. If you have any remaining questions, please do not hesitate to let us know. We are more than happy to have further discussion with you.*

---

### Official Review · Reviewer_hALi · 2025-11-02

**Soundness:** 2
**Presentation:** 2
**Contribution:** 3
**Rating:** 6
**Confidence:** 4

**Summary:**

The manuscript presents the Holistic Protean Block, a DNA language model that is specifically designed to add the capability of identifying structural periodic signals to existing models that are based on CNNs, transformers and state spaces. This capability is implemented in a newly constructed Domain Plasticity Layer that runs parallel to a Saliency Plasticity Layer, and downstream of a Locus Plasticity Layer that tokenizes the input sequence. Results are encouraging, but for some benchmarks inconclusive

**Strengths:**

1. Identificaiton of a systemic weakness in existing models in handling approximate periodicity, and tackling it using the DPL
2. Diverse benchmarking
3. Fig 5a is very convincing in the identification of  agenuine effect

**Weaknesses:**

1. Inconsistency of results
2. Failure to take advantage of existing models that already do much of the required work
3. Lack of clarity regarding parameters
4. Lack of comparison to local models

**Questions:**

1. The authors should separate out the contribution of the DPL (their innovatiive part) from other changes to standard models.(that muddy the water, whether or not they provide some benefit). Table 5 seems to address this, but only on 3 prediction tasks raising the concern of cherrypicking.

2. There are good models for local motifs (the authors cite great models from a decade ago, but progress continued). It is unclear why the authors choose ot have this component redone, rather than stand on the shoulders of giants. and use an existing model. Given that they have not they should compare to such models, esp. if they wish to claim advantage (4 stars vs. 3 in Fig 1). Table 4 seems to try to address this, but only on 3 prediction tasks raising the concern of cherrypicking. Also it compares to strawmen rather than strong local models. Worse, Table 2 shows ConvNova to beat the proposed model by a significant margin on several tasks, whereas when the proposed model beats ConvNova, it is almost always with overlap ping confidence interval and always close to that.

3. I am confused by the changing numbers of parameters of models between tables. If this is due to different lenghts of the sequences, then some sense of consistency should be provided by a unifying formula for this number

4. Fig 5b is unconvincing withiut a comparison to other methods

5. Fig 4 appears after Fig 5.

---

> ### Author Response · Authors · 2025-11-21
> **Part 1 of the responses**
>
> ***We sincerely appreciate your valuable review comments and the effort you have put into them. We have addressed each of the questions you raised in detail. If anything is unclear, please inform us, and we will respond promptly.***
> > Q1. Why Table 5 reports only 3 tasks?
>
> ***Response:*** Due to space constraints in the initial submission, we selected a representative subset of datasets (from the Regulatory, Splice site, and Histone categories) to demonstrate our comparative analysis. To provide a more complete comparison between DPL and other strong methods, we have expanded the ablation study to include all datasets within the Nucleotide Transformer (NT) Benchmark.
> As shown in the results below, our proposed DPL consistently outperforms both 'long conv' (the core module of Hyena) and 'self-attention' (the core mechanism of Transformers) across this entire benchmark. We have included the results in the revised paper (please see A.12) to reflect these comprehensive comparisons.
> Similarly, the complete ablation results for all datasets in the Genomic Benchmark have also been updated and are now presented in Appendix A.11 of the revised paper.
>
> *Comparison of DPL and other methods:*
> | | long conv (Hyena) | self attention (Transformer) | DPL |
> | :--- | :---: | :---: | :---: |
> | Promoter all | 95.73 | 96.02 | **97.01** |
> | Non-TATA | 95.12 | 95.23 | **96.88** |
> | TATA | 96.03 | 96.28 | **96.59** |
> | Enhancer | 53.10 | 52.98 | **58.69** |
> | Enhancer types | 48.66 | 47.98 | **50.87** |
> | Splice Acceptor | 95.99 | 96.36 | **97.16** |
> | Splice Donor | 96.39 | 96.88 | **97.79** |
> | Splice All | 95.82 | 94.11 | **97.12** |
> | H3 | 79.86 | 78.55 | **82.03** |
> | H3K4me1 | 54.02 | 53.11 | **55.39** |
> | H3K4me2 | 51.93 | 53.76 | **53.89** |
> | H3K4me3 | 50.32 | 49.39 | **54.86** |
> | H3K9ac | 60.89 | 59.45 | **64.50** |
> | H3K14ac | 56.91 | 57.87 | **60.55** |
> | H3K36me3 | 62.08 | 62.66 | **63.92** |
> | H3K79me3 | 65.31 | 66.85 | **67.88** |
> | H4 | 78.27 | 79.33 | **83.12** |
> | H4ac | 56.56 | 55.99 | **56.91** |

---

> ### Author Response · Authors · 2025-11-21
> **Part 2 of the responses**
>
> >Q2. Concerns about the comparison to other local motif extractors and ConvNova.
>
> ***Response:*** First, we would like to clarify that we selected the most recent SOTA method (ConvNova [1] 2025) in our comparison of local motif extractors (original Table 4). We compared against its core module, the Gated Convolution Block (GCB), which ConvNova proposes as a highly-optimized **dilated convolution**. To avoid any misunderstanding, we have updated the label from 'dilated conv' to 'GCB' in the revised paper.
> ***Why we design LPL:*** We have now expanded Table 4 (shown below) to include all datasets and several other recent methods, such as MambaOut [2], ConvNeXt [3], and CondConv [4], and have updated the Introduction and Related Works sections accordingly. Compared to LPL, these state-of-the-art local motif extractors still lack sufficient flexibility and adaptability. For example:
> 1. *GCB* (ConvNova) uses relatively static dilated kernels.
> 2. *MambaOut* employs a gated CNN with a fixed kernel.
> 3. *ConvNeXt* uses Depthwise Separable Convolutions with fixed parameters and receptive fields.
> 4. Even highly adaptive methods like *CondConv* only provide **sample-level** adaptation (i.e., all regions within the same sample share the same kernel parameters).
>
> LPL was specifically designed to address this gap by achieving **token-level (or "locus-specific")** dynamics and adaptation. As shown in the updated table below, LPL maintains a clear performance advantage over these state-of-the-art local feature extractors. These results have been included in Appendix A.11 of the revised paper.
> Finally, regarding the comparison with ConvNova in Table 2, our method achieves better average performance across 15 out of 18 tasks. In several key tasks, the performance margin against ConvNova is notable (1.81% on H4, 0.85% on Acceptor, and 0.98% on Enhancer types). Furthermore, our method exhibits consistently higher mean accuracy and lower variance (multiple runs with different random seeds) compared to ConvNova in Table 2, demonstrating **statistically better performance and stability**. Beyond the tasks in Table 2, our model shows a remarkable advantage on the Genomic Benchmark (achieving a nearly 6% gain on the Human Regulatory task) and excels in long-sequence tasks such as Chromatin Profile Prediction and Gene Finding (please see A.15 of the revised paper). Taken together, these results confirm that our method offers a robust advantage in **both accuracy and stability**.
>
> *Comparison of LPL and other strong local motif extractors:*
> || CondConv|ConvNeXt | MambaOut | GCB | LPL |
> | :---:| :---: | :---: | :---:| :---: | :---: |
> | Mouse Enhancers |79.89 | 80.22 | 78.06| 80.38 | **82.38** |
> | Coding vs. Intergenomic |92.11 | 92.38 | 93.19| 93.23 | **94.46** |
> | Human vs. Worm |95.38 | 96.12 | 95.87 |96.29 | **97.19** |
> | Human Enhancers Cohn |73.39 | 72.58 |73.01| 73.03 | **74.96** |
> | Human Enhancers Ensembl | 89.66| 87.85 | 88.03| 87.29 | **91.08** |
> | Human Regulatory | 92.37| 91.99 | 92.96| 91.20 | **94.20** |
> | Human Nontata Promoters | 94.12| 94.20 | 94.13| 94.32 | **94.78** |
> | Human OCR Ensembl | 79.95| 80.81 | 80.35 |80.11 | **81.95** |.
>
> *references*
> [1] Revisiting convolution architecture in the realm of DNA foundation models, ICLR 2025.
> [2] MambaOut: Do We Really Need Mamba for Vision? CVPR 2025.
> [3] A ConvNet for the 2020s, CVPR 2022.
> [4] CondConv: Conditionally Parameterized Convolutions for Efficient Inference, NeurIPS 2019.
>
> >Q3. Why are different model sizes used on different tasks?
>
> ***Response:*** We adopted varying model sizes across different tasks to align with standard configurations in prior works and ensure a fair comparison. For instance, prior works like ConvNova [1], Hyena [2], Caduceus [3], and Bio-LSTM [4] typically utilize models with ~500K parameters for the Genomic Benchmark, ~2M for the Nucleotide Transformer Benchmark, and larger architectures (>6M) for the Chromatin Profile and Variant Effect Prediction tasks.
> Furthermore, our experiments demonstrate that our method achieves better performance even when using comparable or **significantly smaller** model sizes relative to other methods. A notable example is the Chromatin Profile Prediction task, where our model (6.5M parameters) outperforms ConvNova (27.4M parameters) despite being roughly one-quarter the size. This highlights not only the accuracy but also the parameter efficiency of our approach.
>
> *references*
> [1] Revisiting convolution architecture in the realm of DNA foundation models, ICLR 2025.
> [2] HyenaDNA: Long-Range Genomic Sequence Modeling at Single Nucleotide Resolution, NeurIPS 2023.
> [3] Caduceus: Bi-Directional Equivariant Long-Range DNA Sequence Modeling, ICML 2024.
> [4] Bio-xLSTM: Generative modeling, representation and in-context learning of biological and chemical sequences, ICLR 2025.

---

> ### Author Response · Authors · 2025-11-21
> **Part 3 of the responses**
>
> >Q4. Figure 5b is lacking comparison to other methods.
>
> ***Response:*** We wish to clarify that the SAE (Sparse Autoencoder) feature disentanglement visualized in Fig. 5b was primarily intended for interpretability analysis. To provide the requested direct comparison, we conducted additional experiments. Specifically, we pretrained our HPB-DNA model (12.3M) alongside a Transformer (12.6M) and a Hyena model (12.8M), using hg38 DNA sequences with a unified 32k context length. For each model, we trained a corresponding SAE to disentangle its features and evaluated their alignment with genomic functions (promoters, exons, and CDS regions). We utilized the activation of specific sparse features (i.e., non-zero values) to classify distinct genomic functional types (promoters, exons, CDS) and calculated their F1 scores. All models are trained under the same settings. As shown in the table below, under this strictly controlled setting, our model captures genomic information more effectively than the compared methods.
>
> *F1 score between features and genomic regions:*
> |  | Transformer (12.6M) | Hyena (12.8M) | HPB (12.3M) |
> | :--- | :--- | :--- | :--- |
> | promoter | 0.0601 | 0.0900 | **0.1320** |
> | exons | 0.0758 | 0.0830 | **0.0984** |
> | cds| 0.0341 | 0.0500 | **0.0719** |
>
> >Q5. Fig 4 appears after Fig 5.
>
> ***Response:*** We have adjusted the layout in the revised paper to ensure that Figure 4 correctly appears before Figure 5, thanks for reminder.

---

> > ### Comment · Reviewer_hALi · 2025-11-26
> > **Acknowledged responses**
> >
> > The responses and the additional analysis provided adequately address my concenrs.
> > I will update my score

---

> > > ### Author Response · Authors · 2025-11-27
> > >
> > > Thank you so much for your positive feedback and for your willingness to update the score. We also remain fully available and are more than happy to continue the discussion should you have any further inquiries. We deeply appreciate the time and effort you have invested in evaluating our work!

---

### Author Response · Authors · 2025-11-21
**General Response**

We thank all reviewers for their time and positive evaluation of our work. The consensus confirms our goals: our paper addresses "a systemic weakness in existing models" (hALi) and our method "formulating DNA analysis in terms of multi-level plasticity (locus, domain, saliency) is novel" (A7YH). The experiments "are comprehensive" (R4oi), "very convincing" (hALi) and "show consistent improvements under comparable model sizes" (A7YH). Furthermore, our model "surpasses both existing DNA lms and a supervised expert model of Enformer" (oS3Z) and "appears to be a promising way to address genomics tasks" (A7YH).
We also sincerely thank all the reviewers for their insightful comments and suggestions. We have uploaded a revised version of the paper, incorporating the reviewers' helpful suggestions. The main changes are listed below (detailed in the individual responses):
* Clarified the motivation and contribution of each individual module and conducted more comprehensive ablation studies.
* Detailed experimental analysis of model efficiency, covering FLOPs, memory footprint (VRAM), and wall-clock time during training and inference.
* Tested model performance after scaling the architecture to larger parameter sizes (e.g., 100M, 500M).
* Supplemented the missing dataset results in Tables 4 and 5 of the initial submission with complete benchmark results.
* Conducted further biologically interpretable analyses of learned features, using both statistical and visualization methods.
* Added comparisons of the LPL module against more recent and stronger local motif extractors.
* Included new experiments on long-sequence datasets, notably the BEND benchmark.
* Added a related work summary on time/frequency domain signal modeling and clarified/revised specific terminology to enhance readability for a broader audience.

To facilitate the review process, we have marked the revised parts in the paper in blue and provided point-by-point responses to each reviewer's comments. We hope this revised version is clearer, more complete, and demonstrates improved technical justification and empirical validation.
Once again, we extend our gratitude to the reviewers for their efforts and valuable feedback, which have greatly contributed to enhancing the quality of this paper.

---

### Meta-Review · Area_Chair_8fj3 · 2026-01-06

**Summary:**

* The paper introduces a hybrid time-frequency processing layer that captures periodic biological signals like nucleosome positioning.
* The architecture achieves sota results on the BEND benchmark, outperforming existing models by a ~13% margin.
* Token-specific convolutional weights allow the model to adapt to high local variability in DNA motifs better than static filters.
* The model demonstrates strong parameter efficiency, surpassing the Enformer model in variant effect prediction despite being 30 times smaller.
* Experimental results show the architecture scales effectively to 500M parameters with consistent performance gains.
* Computational complexity is quasi-linear, maintaining efficiency on sequences up to 131k base pairs.

This paper presents the "Holistic Protean Block", a neural network architecture for DNA sequence modeling. The core innovation is the integration of a wavelet-based spectral path alongside global convolutions. This allows the model to identify periodic genomic patterns that are often invisible to standard time-domain architectures. The model also utilizes adaptive local filters and dual-axis gating to manage information flow across long sequences.

The method performs well across short-range classification + ultra-long-range prediction tasks. The authors demonstrated that the architectural choices (eg. frequency-domain analysis) are well adapted to tackle known biological properties. While the performance boost on some standard benchmarks is limited, the efficiency and interpretability of the spectral features provide significant value.

**Reviewer Concerns:**

**Addressed by rebuttal**

* [Core] Efficiency verification: The authors provided empirical FLOPs and runtime data to support their theoretical complexity claims. Resolved.
* [Core] Model scalability: New experiments on 100M and 500M parameter versions demonstrated clear scaling laws for the architecture. Resolved.
* [Core] Local modeling baselines: Comparisons with recent adaptive convolutional methods showed the benefit of token-specific weights. Resolved.
* [Core] Biological justification: The link between the wavelet path and nucleosome periodicity was clarified with spectral density analysis. Resolved.
* [Non-core] Terminology clarity: Definitions for domain-specific terms like locus plasticity and meta-features were improved for readability. Resolved.

**Still outstanding**

* [Core] Statistical significance: Performance margins on the Nucleotide Transformer benchmark remain small relative to standard deviations. Partially resolved.
* [Core] Pre-training constraints: The primary model comparison still relies on 1k-length pre-training, which may limit the assessment of long-range dependencies. Partially resolved.

The rebuttal period was productive. The authors addressed major gaps regarding empirical efficiency + model scaling. The addition of the BEND benchmark strengthened the evidence for long-range modeling. Some concerns about the magnitude of improvement on specific benchmarks remain.

**Reviewer Scores:**

* **Reviewer hALi**
* Original score: 6
* Estimated score shift: increase
* Reviewer explicitly noted that the additional analyses and spectral evidence adequately addressed their technical concerns.

* **Reviewer A7YH**
* Original score: 4
* Estimated score shift: increase
* Reviewer acknowledged the resolution of concerns regarding scaling laws, efficiency benchmarks, and biological motivation.

* **Reviewer R4oi**
* Original score: 2 (--> 4)
* Estimated score shift: increase
* After initially providing a very low score, the reviewer increased their assessment to 4 during the first rebuttal phase [not seen here, but see discussion]. They conceded that their initial critique of the "long-range" claims was based on a misunderstanding. While still skeptical about the modular design, the primary objections regarding pre-training lengths and definitions were resolved by the authors' evidence.

* **Reviewer oS3Z**
* Original score: 4
* Estimated score shift: unchanged
* This reviewer ** did not participate in the discussion **, though their requests for throughput analysis and harder benchmarks were fulfilled in the author's general responses.

The review panel moved toward a positive consensus during the discussion. Two reviewers indicated that their major technical hurdles were cleared. One reviewer remained critical based on subjective benchmark preferences, but their factual errors regarding model capacity and pre-training were corrected by the authors.

---

### Decision · Program_Chairs · 2026-01-26

Reject